# Delineation of G Protein-Coupled Receptor Kinase Phosphorylation Sites within the D_1_ Dopamine Receptor and Their Roles in Modulating β-Arrestin Binding and Activation

**DOI:** 10.3390/ijms24076599

**Published:** 2023-04-01

**Authors:** Amy E. Moritz, Nora S. Madaras, Michele L. Rankin, Laura R. Inbody, David R. Sibley

**Affiliations:** Molecular Neuropharmacology Section, National Institute of Neurological Disorders and Stroke, National Institutes of Health, 35 Convent Drive, Bethesda, MD 20892, USA

**Keywords:** dopamine, D_1_ receptor, phosphorylation, β-arrestin

## Abstract

The D_1_ dopamine receptor (D1R) is a G protein-coupled receptor that signals through activating adenylyl cyclase and raising intracellular cAMP levels. When activated, the D1R also recruits the scaffolding protein β-arrestin, which promotes receptor desensitization and internalization, as well as additional downstream signaling pathways. These processes are triggered through receptor phosphorylation by G protein-coupled receptor kinases (GRKs), although the precise phosphorylation sites and their role in recruiting β-arrestin to the D1R remains incompletely described. In this study, we have used detailed mutational and in situ phosphorylation analyses to completely identify the GRK-mediated phosphorylation sites on the D1R. Our results indicate that GRKs can phosphorylate 14 serine and threonine residues within the C-terminus and the third intracellular loop (ICL3) of the receptor, and that this occurs in a hierarchical fashion, where phosphorylation of the C-terminus precedes that of the ICL3. Using β-arrestin recruitment assays, we identified a cluster of phosphorylation sites in the proximal region of the C-terminus that drive β-arrestin binding to the D1R. We further provide evidence that phosphorylation sites in the ICL3 are responsible for β-arrestin activation, leading to receptor internalization. Our results suggest that distinct D1R GRK phosphorylation sites are involved in β-arrestin binding and activation.

## 1. Introduction

The dopaminergic system regulates numerous physiological processes in both the central nervous system and periphery, including cognition, mood, movement, reward, as well as cardiovascular and renal physiology. Aberrant dopaminergic regulation contributes to diverse diseases including Parkinson’s disease, schizophrenia, substance use disorder, and hypertension. Dopaminergic signaling is mediated by five G protein-coupled receptors (GPCRs) that are subdivided into two families based on sequence homology, signaling pathways, and pharmacological profiles. The D1-like dopamine receptors (DRs) consist of the D1R and D5R, which couple to Gαs/olf to activate adenylyl cyclase and increase cAMP levels, whereas the D2-like receptors, consisting of the D2R, D3R, and D4R, functionally couple to Gαi/o to inhibit adenylyl cyclase activity [1,2,3].

In addition to G protein-mediated signaling, all activated GPCRs recruit scaffolding proteins from the arrestin family that consists of four members—arrestins 1 and 4 that are restricted to the visual system, and arrestins 2 and 3, more commonly referred to as β-arrestin 1 and 2, respectively, which are ubiquitously found in most cell types [4]. Recruitment of β-arrestin initiates receptor desensitization and internalization [5], but is also recognized to independently activate distinct downstream signaling cascades [4,6]. The desensitization process initiated by β-arrestin recruitment is typically linked to phosphorylation of GPCRs by serine/threonine protein kinases [7,8,9]. Homologous desensitization and β-arrestin recruitment involve phosphorylation of agonist-activated receptors by members of the G protein-coupled receptor kinase family (GRKs) [10,11], whereas GPCRs can also be desensitized in a heterologous fashion through phosphorylation by second-messenger-activated kinases, such as protein kinase A (PKA) and protein kinase C (PKC) [12].

The D1R possesses ~32 intracellular serine and threonine residues and appears to be phosphorylated by multiple protein kinases. We [13,14,15], and others [16], have shown that the D1R is phosphorylated by multiple PKC isozymes and that this mainly occurs constitutively and dampens G protein coupling and signaling. Notably, in HEK293 cells, most of the D1R phosphorylation observed in the basal state is mediated by PKCs, which can be reversed by treatment with PKC-selective inhibitors [14,16] or recreational concentrations of ethanol [13]. Importantly, ethanol treatment was shown to potentiate D1R-mediated DARPP-32 phosphorylation in rat striatal slices, demonstrating that ethanol can enhance D1R signaling in vivo. Our laboratory has mapped the PKC-mediated phosphorylation sites in the rat D1R to residues Ser-259, Ser-397, Ser-398, Ser-417, and Ser-421 in both the third intracellular loop (ICL3) and C-terminus (Figure 1) [14]. Interestingly, residue Ser-421 in the C-terminus (S14 in Figure 1) has been reported to also serve as a substrate for protein kinase D1 (PKD1) and that PKD1-mediated phosphorylation may enhance D1R cell surface expression and regulate cocaine-induced behavioral responses [17,18]. Notably, this residue has also been shown to be important for D1R interactions with glycogen synthase kinase-3 (GSK-3) [19]. Other second-messenger-activated kinases that are known to phosphorylate the D1R include the cAMP-dependent protein kinase (PKA), which phosphorylates a single residue, Thr-268. PKA-mediated phosphorylation appears to regulate the rate of agonist-induced desensitization [20] and/or receptor trafficking [21,22].

The D1R is also known to be phosphorylated in response to agonist activation [23,24,25,26,27,28,29], which is mediated by GRKs and associated with receptor desensitization and endocytosis. Whereas GRK2, GRK3, and GRK5 were found to phosphorylate the D1R in an agonist-dependent fashion [23,26,27,28], D1R phosphorylation by GRK4 (specifically the GRK4α isoform) was found to be independent of agonist stimulation, or constitutive in nature [28]. Notably, constitutive hyper-phosphorylation of the D1R by GRK4α in the kidney has been hypothesized to promote elevated blood pressure and may be linked to forms of hypertension [30,31,32]. Our laboratory has found that constitutive phosphorylation by GRK4α takes place on five serine and threonine residues (Thr-428, Ser-431, Ser-441, S445, and Thr-446) in the distal C-terminus of the D1R (Figure 1) and is associated with receptor desensitization and internalization [28].

The location of other GRK-mediated phosphorylation sites within the D1R has proven to be controversial. Lamey et al. (2002) suggested that dopamine activation promotes the phosphorylation of a single residue, Thr-360, in the proximal C-terminus of the D1R (T5 in Figure 1) resulting in receptor desensitization [26]. In contrast, through examining a series of truncation mutants, the Tiberi group has suggested that GRK2/3-mediated phosphorylation of the D1R occurs on multiple residues within the central and distal regions of the C-terminus [25,29]. Our laboratory also investigated potential sites of agonist-stimulated GRK phosphorylation within the D1R [27]. We found that progressive truncation of the C-terminus resulted in diminished agonist-stimulated D1R phosphorylation, indicating that multiple GRK sites (other than those phosphorylated by GRK4α) exist in the C-terminus of the D1R. However, we also found evidence for agonist-induced phosphorylation of a cluster of three serine residues in the ICL3 (S256, S258, and S259), which further regulate β-arrestin translocation to the D1R. Surprisingly, truncating most of the C-terminus of the D1R completely negated the ability of the receptor to be phosphorylated in response to agonists, suggesting that the presence of the C-terminus can modulate phosphorylation of GRK sites in the ICL3 [27]. In the current study, we have used detailed site-directed mutational analyses to completely identify the agonist-stimulated, GRK-mediated phosphorylation sites on the D1R. Our results indicate that GRK-mediated phosphorylation of residues on the C-terminus and ICL3 occurs in a hierarchical fashion and that phosphorylation sites in the proximal region of the C-terminus are critical for β-arrestin binding, whereas phosphorylation sites in the ICL3 are mainly responsible for β-arrestin-mediated receptor internalization.

**Figure 1 ijms-24-06599-f001:**
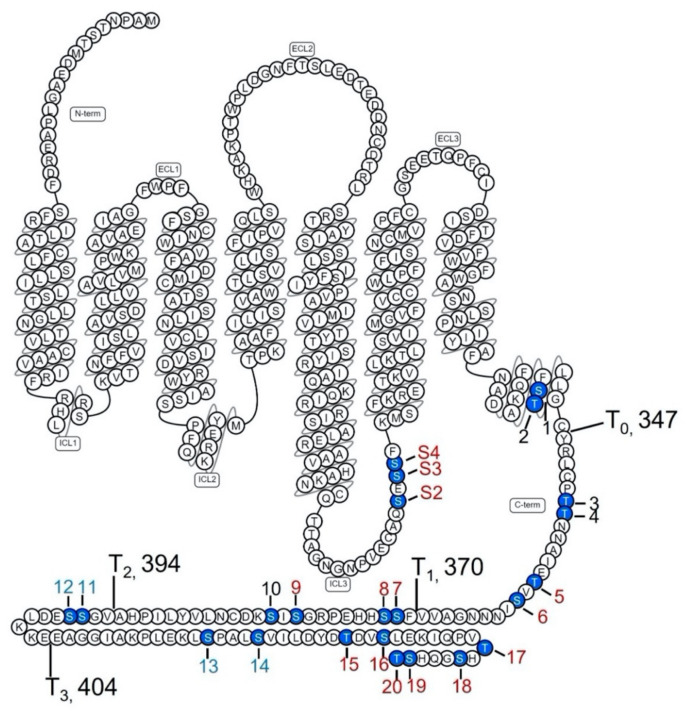
Diagram of the rat D1R illustrating the receptor mutants used in this study. Blue filled circles indicate serine and threonine residues that were mutated in the ICL3 and C-terminal tail. Four carboxyl terminal truncation mutants were also generated by inserting stop codons after amino acids 347 (T_0_), 370 (T_1_), 394 (T_2_), or 404 (T_3_). The T_0_ truncation site was based on the observation that Cys-347 in the rat D1R is palmitoylated. The twenty serine and threonine residues in the C-terminus are numbered 1–20. Simultaneous mutation of these 20 residues refers to the “Tail Total” mutant. Simultaneous mutation of the eight serine and threonine residues in the distal C-terminus refers to the “S13–20” mutant. Simultaneous mutation of just the three Ser residues (S256, S258, and S259) in the ICL3 refers to the “S2/S3/S4” mutant. The GRK-mediated D1R phosphorylation sites determined in this study, as well as in Rankin et al. (2006) [28] are delineated using red numbering and lettering on the receptor diagram. Simultaneous mutation of these residues refers to the “GRK-null” mutant. The sites that are constitutively phosphorylated by GRK4α are: T15, S16, S18, S19 and T20 [28]. PKC-mediated phosphorylation sites that were determined in Rankin and Sibley, 2010 are shown in blue numbering (11–14) on the receptor diagram. Note that residue S4 (S259) in the ICL3 has been identified as being phosphorylated by both GRKs and PKCs [14,27]. The D1R snake diagram was modified from GPCRdb.org [33].

## 2. Results

### 2.1. Determination of GRK-Mediated Phosphorylation Sites within the Rat D1R

We had previously identified the D1R residues that are phosphorylated by PKCs by utilizing a combination of C-terminal truncation mutants as well as almost 100 different point mutants (see blue numbered residues in Figure 1) [14]. In the current study, we set out to completely delineate the residues in the D1R that are phosphorylated by GRKs upon DA stimulation. As an approach to narrow down the location of residues phosphorylated by GRKs, we examined a series of C-terminal truncation mutants and observed that progressive truncation decreased the ability of the D1R to be phosphorylated upon DA stimulation (Figure 2a). The most severe truncation, T_0_, lacks the ability to be phosphorylated, recapitulating our previous findings [14] and those of Jackson et al. (2002) using a similar mutant [25]. Since the D1R C-terminus appears to be heavily phosphorylated, a reverse mutational approach was next used in which all 20 potential phosphorylation sites in this region were mutated from serines and threonines to alanines and valines, respectively (Figure 1). This mutant construct, referred to as “Tail Total”, provides a null background in which, notably, there is no DA-induced phosphorylation of the D1R (Figure 2b). Interestingly, mutating the previously identified [27] cluster of three serines, S2/S3/S4 (S256, S258, and S259), in the D1R ICL3 (Figure 1) reduces D1R phosphorylation in response to DA stimulation (Figure 2a), suggesting that these are GRK sites; however, these residues are not phosphorylated within the context of either the T_0_ mutant (Figure 2a) or the Tail Total mutant (Figure 2b) indicating that phosphorylation of the C-terminus is required for these ICL3 residues to be phosphorylated. In other words, it appears that the D1R undergoes hierarchical phosphorylation by GRKs such that phosphorylation of the C-terminus enables phosphorylation of the ICL3 residues.

To further identify specific residues or clusters of GRK phosphorylation sites in the D1R, we next created “reverse” (Rev) mutants where residues in the Tail Total were mutated back to their WT sequence (serine or threonine) and phosphorylation assays were performed. We found that the Rev S1T2 (S342 and T343) mutant was not phosphorylated upon DA stimulation (Figure 2c), suggesting that these residues are not GRK sites. These data are in agreement with those found for the T_0_ truncation mutant, which lacks DA-induced phosphorylation (Figure 2a) while it retains the WT S1 and T2 residues (Figure 1). The Rev T3T4 (T353 and T354) mutant also lacks DA-induced phosphorylation, arguing that these residues are not GRK-mediated phosphorylation sites (Figure 2c). In contrast, the Rev T5S6 (T360 and S362) mutant partially rescued DA-induced phosphorylation (30 ± 2.6% vs. WT, mean ± SEM) (Figure 2c) suggesting that these are GRK sites. The Rev S7S8 (S372 and S373) construct displayed a low level of phosphorylation upon DA stimulation that was only visible upon over-expression of the film (Figure 2d), suggesting that these residues may be minor GRK phosphorylation sites (however, see below). The Rev S9S10 (S380 and S382) mutant also gained an ability to be phosphorylated upon DA stimulation (19 ± 1.8% vs. WT, mean ± SEM) (Figure 2e), suggesting that these residues are also sites of GRK phosphorylation. However, a construct containing only Rev S10 displayed no DA-induced phosphorylation (Figure 2e), indicating that S9 is the sole GRK phosphorylation site in the Rev S9S10 construct. A Rev S7–S10 construct displayed levels of DA-induced phosphorylation similar to that of the Rev S9S10 construct (24 ± 2.7% vs. WT, mean ± SEM), again suggesting that S9 is the main site of GRK phosphorylation within this cluster of residues (Figure 2e).

### 2.2. Overexpression of GRKs Can Enhance DA-Induced Phosphorylation of Specific D1R Residues

As we [28], and others [23], have found that overexpression of GRK2 or GRK3 can enhance DA-induced phosphorylation of the D1R, we tested a subset of reverse D1R mutants to see if overexpression of either GRK2 or GRK3 could either promote DA-induced phosphorylation of residues not visualized under conditions of endogenous GRK expression levels or increase the phosphorylation states of residues that we determined to be GRK phosphorylation sites. The Rev T3T4 mutant, which was not phosphorylated by DA stimulation in nontransfected cells, also failed to be phosphorylated with either GRK2 or GRK3 overexpression (Figure 3). In contrast, the Rev S7S8 mutant, which displayed a low level of phosphorylation in response to DA stimulation under endogenous GRK levels, displayed enhanced phosphorylation upon GRK2 or GRK3 overexpression, suggesting that these residues may indeed be sites of significant GRK phosphorylation.

We have previously found that expression of GRK4α in HEK293 cells, which is not normally expressed in these cells [34], can lead to constitutive phosphorylation of serine and threonine residues in the distal C-terminus of the D1R, with T15 (T428) and S16 (S431) representing the main sites [28]. In order to determine if these sites are unique to GRK4 or can also be phosphorylated by other GRKs, we tested if overexpression of GRK2 or GRK3 would promote phosphorylation of reverse mutants in this distal Ser/Thr cluster. We found that the Rev T15S16 (T428 and S431), Rev T17S18 (T439 and S441), and Rev S19T20 (S445 and T446) mutants all display DA-induced phosphorylation when GRK2 or GRK3 are overexpressed (Figure 3), indicating that these sites can be phosphorylated by multiple GRKs with GRK2 and GRK3 requiring an activated receptor. Taken together, our current and previous data [27,28] suggest that the residues highlighted with red lettering in Figure 1 represent the GRK phosphorylation sites in the rat D1R.

### 2.3. GRK Phosphorylation Sites in the D1R C-Terminus Are Critical for β-Arrestin Recruitment

Having identified the residues in the D1R that are phosphorylated in response to receptor activation by DA, we were next interested in characterizing the functional effects resulting from this phosphorylation. We initially examined the ability of the C-terminal truncation mutants to recruit β-arrestin by utilizing the CAAX β-arrestin recruitment assay [35]. We found that the most severe truncation, D1R T_0_, which lacks DA-induced phosphorylation, also displays a severely impaired ability to recruit β-arrestin (Emax = 16 ± 4.4% vs. WT D1R, *p* < 0.0001) (Figure 4a). Since we previously found, using live-cell microscopy [27], that this mutant construct, when stimulated with DA, promoted subcellular translocation of GFP-tagged β-arrestin, we turned to an orthogonal β-arrestin recruitment BRET assay (Appendix A), and, similar to the CAAX β-arrestin assay results, we found that the D1R T_0_ mutant displays severely impaired DA-induced β-arrestin recruitment (Emax = 13 ± 6.5% vs. WT, *p* = 0.0002). We are not certain why different results were observed using the microscopy-based assay [27], but it might involve heterogeneous expression of the assay plasmids and difficulty in quantifying single cell data using microscopy. In contrast, the BRET assays used in this study quantify the results from the entire population of cells in the assay plate and are more reliable and quantitative. Notably, the D1R T_1_ truncation mutant, which contains the T5 and S6 residues, recruits β-arrestin to the same extent as WT (Emax = 110 ± 9.3% vs. WT, *p* = 0.5744). These results indicate that the GRK phosphorylation of residues T5 and S6 play a central role in DA-induced β-arrestin recruitment to the D1R.

The D1R T_2_ truncation mutant, which, in addition to residues T5 and S6, also contains GRK phosphorylation sites S7, S8, and S9, recruits β-arrestin to the same extent as WT (Emax = 90 ± 10% vs. WT, *p* = 0.5726). These data further suggest that residues T5 and S6 are primarily involved in β-arrestin recruitment. Interestingly, we found that the D1R T_3_ truncation mutant, which is missing the distal part of the C-terminus, was able to recruit β-arrestin to an even greater extent than WT (Emax = 150 ± 0.6% vs. WT, *p* = 0.0003). This could suggest that the distal C-terminus of the D1R may, in some way, negatively affect D1R-β-arrestin interactions. To determine if either removal of the distal C-terminus or elimination of the phosphorylation sites within this region explains the enhanced β-arrestin recruitment to the D1R T_3_ mutant, we evaluated β-arrestin recruitment using a D1R mutant in which the C-terminus is intact, but the eight serine and threonine residues, S13–S20, in this region were mutated to alanine and valine, respectively. As shown in Appendix A, this mutant construct, D1R S13–S20, exhibited impaired β-arrestin recruitment compared to WT (Emax = 78 ± 1.7% vs. WT, *p* = 0.0002), although G protein coupling was unaffected (Appendix A). These results suggest that the distal C-terminus may physically modulate β-arrestin recruitment to the D1R, although the exact mechanism is unclear. More importantly, these results suggest that GRK phosphorylation sites within the distal C-terminus may partially contribute to β-arrestin recruitment (also, see below). In contrast to the effects of the D1R mutations on β-arrestin recruitment, all mutant constructs exhibited G protein signaling that was indistinguishable from the D1R WT (Figure 4b and Appendix A).

### 2.4. Identification of Individual GRK Phosphorylation Sites That Are Responsible for β-Arrestin Recruitment

In order to evaluate the contributions of individual phosphorylation sites on DA-induced β-arrestin recruitment to the D1R, we initially evaluated the ICL3 mutant in which the cluster of three serine residues, S2/S3/S4 (Figure 1), are mutated to alanines. We found that these mutations partially impaired the ability of the receptor to recruit β-arrestin using the CAAX BRET assay (Emax = 61 ± 0.8% vs. WT, *p* < 0.0001) (Figure 4c), although there was no effect on G protein interactions (Figure 4d). These results suggest that phosphorylation of these residues contribute to D1R-β-arrestin interactions and recapitulate our previous findings using a different β-arrestin recruitment assay [27]. To evaluate the residues in the C-terminus, we started with the phosphorylation-null D1R Tail Total construct (Figure 1). We found that this mutant construct, in which all 20 potential phosphorylation sites in the C-terminus are mutated to alanine and valine, is severely impaired in DA-stimulated β-arrestin recruitment (Figure 4c). We then tested reverse (Rev) mutants containing clusters of serine and/or threonine residues that were added back to the D1R Tail Total mutant. Notably, the following reverse mutants were unable to rescue DA-induced β-arrestin recruitment to any extent: Rev S1T2, Rev T3T4, Rev S7S8, Rev S9S10, Rev S7–S10 in the proximal C-terminus (Appendix A) or Rev T15S16, Rev S17S18, or Rev S19T20 in the distal C-terminus (Appendix A), even though we found that the Rev S7S8, Rev S9S10, Rev T15S16, Rev T17S18, and Rev S19T20 constructs could be phosphorylated in response to DA stimulation (Figure 2 and Figure 3). These D1R mutants exhibited normal G protein interactions (Appendix A) except for the Rev S7–S10 and Rev S17S18 mutants, which each exhibited a ~2-fold (*p* < 0.05) increase in the DA EC_50_ values (Appendix A). In contrast, the Rev T5S6 mutant was able to partially rescue DA-stimulated β-arrestin recruitment (Figure 4C). The extent of β-arrestin recruitment to the Rev T5S6 mutant (Emax = 46 ± 4.4% vs. WT, *p* < 0.0001) was comparable to its level of DA-stimulated receptor phosphorylation (Figure 2c). Unlike the effects of these mutations on β-arrestin recruitment, they all displayed little to no effect on G protein signaling, although the Tail Total and S2/S3/S4 mutants exhibited statistically significant increases (~10%) in the DA Emax values (*p* < 0.05) and the DA EC_50_ value for the S2/S3/S4 mutant was increased by ~3-fold (*p* < 0.05) (Figure 4D).

Because the data with both the truncation mutants and the reverse Tail Total mutants indicate that residues T5 and S6 play a central role in DA-induced β-arrestin recruitment to the D1R, we next evaluated the mutation of these residues within the context of the WT D1R. We found that a D1R T5V/S6A mutant exhibits impaired DA-stimulated β-arrestin recruitment (Figure 5a) while having no effect on G protein interactions (Figure 5b). We additionally mutated these residues on the background of the D1R T_1_ truncation mutant, as this construct has normal β-arrestin recruitment (Figure 4a) and contains residues T5 and S6 (Figure 1). As shown in Figure 5c, the D1R T_1_ + T5V/S6A mutant also exhibits impaired DA-induced β-arrestin recruitment (Emax = 17 ± 10.5% vs. WT, *p* = 0.0002), while having no effect on G protein interactions (Figure 5d).

To further investigate the role of residues T5 and S6 on β-arrestin recruitment to the D1R, we utilized a direct D1R-β-arrestin recruitment BRET assay in which mVenus-tagged β-arrestin is recruited to an Rluc8-tagged receptor. As shown in Figure 6a, DA-stimulated β-arrestin recruitment was impaired in the D1R Tail Total mutant (Emax = 25 ± 3.1% vs. WT, *p* < 0.0001), although not to the extent observed with the CAAX β-arrestin assay (Figure 4c). Notably, β-arrestin recruitment to the D1R T5V/S6A mutant was similarly impaired (Emax = 36 ± 4.1% vs. WT, *p* < 0.0001) (Figure 6b), consistent with the idea that phosphorylation of these residues is critically involved in β-arrestin recruitment to the D1R. To test whether mutation of one of these residues causes a greater effect on β-arrestin recruitment than the other, we evaluated β-arrestin recruitment using single D1R T5V and D1R S6A point mutants (Figure 6c). Interestingly, mutation of either residue resulted in a similar blunting of β-arrestin recruitment (D1R T5V Emax = 39 ± 5.4% vs. WT, *p* < 0.0001; D1R S6A Emax = 39 ± 2.8% vs. WT, *p* < 0.0001) (Figure 6c) to the same level as that seen with the double mutant (Figure 6b). These data suggest that phosphorylation of both residues in this cluster is necessary to fully drive β-arrestin recruitment to the D1R.

Notably, the degree of β-arrestin recruitment in the forward T5V/S6A mutant (Emax = 36 ± 4.1% vs. WT, *p* < 0.0001) (Figure 6b) is similar to that seen with the reverse Tail Total T5S6 mutant (Emax = 46 ± 4.4% vs. WT, *p* < 0.0001) (Figure 4c). Our interpretation of these results is that while GRK phosphorylation of the T5S6 cluster plays a central role in D1R-β-arrestin interactions, phosphorylation of other residues, such as the ICL3 S2/S3/S4 cluster, is required for maximal β-arrestin recruitment to the D1R. Further, while GRK phosphorylation of C-terminal residues, other than T5S6, is not sufficient to drive β-arrestin recruitment to the D1R (Appendix A), phosphorylation of these additional residues in the C-terminus in conjunction with T5S6 may enhance D1R-β-arrestin interactions or, perhaps more interestingly, play a role in the hierarchical phosphorylation of the ICL3 serine cluster, or even the T5S6 cluster, which might explain the slightly decreased β-arrestin recruitment seen with the D1R S13–20 mutant (Appendix A).

### 2.5. Mutation of the D1R GRK-Mediated Phosphorylation Sites Eliminate β-Arrestin Recruitment

Based on our analyses above, we have identified D1R residues T5, S6, S7, S8, S9, T15, S16, T17, S18, S19, and T20 in the C-terminus, and residues S2, S3, and S4 in the ICL3 as being phosphorylated by GRKs upon stimulation with DA. We thus created a “GRK-null” D1R mutant in which these residues were simultaneously mutated to either alanines or valines and investigated the aggregate effects of these mutations on D1R function. Using both the CAAX β-arrestin BRET assay (Figure 7a) and the direct D1R-β-arrestin recruitment BRET assay (Figure 7b), we found that the D1R GRK-null receptor completely failed to recruit β-arrestin upon DA stimulation, indicating that phosphorylation of these sites is required for β-arrestin recruitment to the D1R. We did notice, however, that this mutant exhibited a slightly reduced potency for DA in the cAMP accumulation assay (Figure 7c), which may be due to its reduced expression level (Appendix A) or due to a direct effect on receptor-G protein interactions. Taken together, these results indicate that GRK-mediated receptor phosphorylation is necessary for DA-stimulated β-arrestin recruitment to the D1R.

### 2.6. D1R Residues Phosphorylated by PKC Have Minimal Effects on Dopamine-Stimulated β-Arrestin Recruitment

As noted above, our laboratory previously identified D1R residues S11 (S397), S12 (S398), S13 (S417), and S14 (S421) in the C-terminus, and residue S4 (S259) in the ICL3 (Figure 1) as being sites of constitutive PKC phosphorylation [14]. Notably, we determined that most of the basal D1R phosphorylation is mediated by PKCs and that this dampens D1R function [13,14,15]. Because we have now delineated the residues phosphorylated by GRKs and found them to be almost exclusively separate from those phosphorylated by PKCs, with the exception of the ICL3 S4 serine, we wanted to test if these PKC-phosphorylated residues play any role in modulating β-arrestin recruitment to the D1R. We initially tested a D1R PKC-null mutant in which serines S11, S12, S13, and S14 in the C-terminus and S4 in the ICL3 were simultaneously mutated to alanines. Figure 8a shows that DA-stimulated β-arrestin recruitment to this PKC-null mutant is somewhat blunted (Emax = 71 ± 3.5% vs. WT, *p* = 0.0004) suggesting that these residues may have some influence on β-arrestin recruitment. As this D1R PKC-null construct has S4 (S259) in the ICL3 mutated to alanine, and we found that this residue is phosphorylated by GRKs, we mutated this residue back to serine in the D1R PKC-null construct to see if this would increase β-arrestin recruitment. However, as shown in Figure 8a, the A259S mutation within the background of the D1R PKC-null construct had no appreciable effect on DA-stimulated β-arrestin recruitment (D1R PKC-null + A259S Emax = 59 ± 9.4% vs. WT, *p* < 0.0001) compared to the PKC-null construct, suggesting that phosphorylation of this residue alone is insufficient to drive β-arrestin recruitment. Notably, these D1R mutants were not impaired in their G protein interactions (Figure 8b).

We next tested a reverse PKC-null mutant (D1R Rev PKC), in which serines S11, S12, S13, and S14 in the C-terminus were reverted to WT within the D1R Tail Total mutant (note that S4 (S259) in this construct is WT), in which β-arrestin recruitment is severely impaired (Figure 4), to see if restoring the PKC phosphorylation sites could partially rescue DA-stimulated β-arrestin recruitment. However, Figure 8c shows that the D1R Rev PKC mutant is incapable of DA-stimulated β-arrestin recruitment, although G protein interactions are normal (Figure 8d). Our interpretation of these data is that basal PKC-mediated phosphorylation of the D1R might enhance β-arrestin recruitment to the receptor once it is phosphorylated by GRKs, thus explaining the slight decrease in β-arrestin recruitment seen in the PKC-null mutants (Figure 8a), but that PKC-mediated phosphorylation alone cannot initiate β-arrestin recruitment as demonstrated by the D1R Rev PKC mutant (Figure 8c). In fact, these data are reminiscent of those observed with the D1R S13–20 mutant, which exhibits a slight impairment in β-arrestin recruitment (Appendix A), but restoration of these mutated GRK phosphorylation sites alone is insufficient to restore β-arrestin recruitment to the D1R Tail Total mutant (Appendix A). Taken together, our data suggest that while GRK-mediated phosphorylation of the T5S6 cluster is critical for DA-stimulated β-arrestin recruitment, other GRK-mediated phosphorylation sites play either primary (i.e., the ICL3 serine cluster) or secondary roles (i.e., other C-terminal clusters) and that even PKC-mediated phosphorylation may exert modulatory effects.

### 2.7. GRK-Mediated Phosphorylation of the S2/S3/S4 Cluster in ICL3 Is Necessary for DA-Stimulated D1R Internalization

As our lab has previously shown that DA-stimulated D1R internalization is a β-arrestin-mediated process [28], we next sought to determine how receptor internalization is affected in mutants that display defects in their ability to be phosphorylated. Utilizing a Lyn-kinase proximity BRET internalization assay [35], we found that the D1R Tail Total mutant, which lacks GRK-mediated phosphorylation (Figure 2b) and does not recruit β-arrestin upon dopamine stimulation (Figure 4c), exhibits severely blunted receptor internalization in comparison to the WT D1R (Figure 9). Similar results were observed for the GRK-null construct, which is unable to recruit β-arrestin (Figure 7) and exhibits significantly impaired DA-stimulated D1R internalization (Figure 9). Surprisingly, the T5V/S6A mutant, which exhibits partial phosphorylation (Figure 2c) and partial β-arrestin recruitment (Figure 5 and Figure 6), displays DA-stimulated D1R internalization that is close to that of the WT D1R (Figure 9), suggesting that other GRK sites must be necessary for D1R internalization. As we have shown that the Tail Total mutant lacks phosphorylation not only on the C-terminus, but also on the S2/S3/S4 serine cluster in the ICL3, we tested the ability of the D1R S2/S3/S4 mutant to internalize upon DA stimulation. Interestingly, we found that DA-stimulated internalization of the S2/S3/S4 mutant was impaired to a similar degree as the GRK-null and Tail Total mutants (Figure 9). These data suggest that the impaired internalization of the latter D1R mutants is due to the inability of the S2/S3/S4 cluster in ICL3 to be phosphorylated in these constructs. These data further suggest that while D1R β-arrestin recruitment and receptor internalization are intimately linked, these processes are controlled through the phosphorylation of distinct residues within the D1R.

## 3. Discussion

In this study, we have endeavored to completely map the GRK-mediated phosphorylation sites on the D1R and begin an assessment of their specific roles in regulating receptor function. Multiple studies have previously described agonist-stimulated D1R phosphorylation (see Section 1); however, a complete map of the GRK phosphorylation sites has remained an elusive goal for several reasons. Firstly, the cytoplasmic surface of the D1R contains approximately 32 serine and threonine residues and appears to be heavily phosphorylated. Secondly, the D1R is phosphorylated by multiple protein kinases including PKA, PKCs, PKD1, GRKs, and perhaps others, with some overlap in substrate specificity. Thirdly, the D1R is constitutively phosphorylated by PKCs in HEK293 cells, thus increasing the background level of phosphorylation. Fourthly, D1R phosphorylation by GRKs appears to occur in a hierarchical fashion such that phosphorylation of some residues is required for the phosphorylation of other residues. To accomplish our goal of identifying all GRK sites, we used several approaches, including both truncation and site-directed mutagenesis. One important approach was our use of a phosphorylation-null D1R construct in which all serine and threonine residues in the C-terminus were first mutated to alanines or valines (the Tail Total mutant), which was then used to revert specific residues to their wild-type status to detect if this resulted in any gain of D1R phosphorylation. The clean phosphorylation-null background of the Tail Total mutant thus allowed for a more sensitive method of detecting both individual and clusters of phosphorylation sites within the D1R. Taking our current, and previous [27,28], mutational and in situ phosphorylation data together, we propose that the D1R possesses 14 serine and threonine residues in its ICL3 and C-terminus that can be phosphorylated by GRKs in response to agonist stimulation (Figure 1, red numbering).

Interestingly, it appears that GRK-mediated phosphorylation of the D1R is partially hierarchical in nature. This was first suggested by observations showing that near complete truncation of the C-terminus lead to a complete loss of receptor phosphorylation [14,25,27]. However, in our previous study [27], we found that mutation of the serine and threonine residues in the ICL3 of the WT D1R led to the identification of a cluster of serines, S2/S3/S4 (S256, S258, and S259) that are phosphorylated in response to agonist stimulation. These results suggested that the intact C-terminus is required for the ICL3 to be phosphorylated, although we could not distinguish between a hierarchical mechanism, where phosphorylation of C-terminal residues must first take place for subsequent phosphorylation of ICL3 residues, or if the C-terminus was simply needed to be physically present, perhaps for GRK docking or orientation. Our results with the Tail Total mutant clearly argue in favor of the hierarchical phosphorylation hypothesis, although the specific mechanism for how phosphorylation of the C-terminus leads to phosphorylation of the ICL3 is unclear. Previously, we speculated that phosphorylation of the C-terminus might prevent it from sterically hindering ICL3 phosphorylation by GRKs [27]. Interestingly, there are studies showing that functional interactions exist between the D1R ICL3 and the C-terminus in support of this hypothesis [36,37], although more detailed experiments will be required to elucidate the precise mechanism. Notably, hierarchical phosphorylation has been previously described for a number of different GPCRs [38,39,40,41].

Since the major role of GRK phosphorylation is to drive or enhance β-arrestin recruitment to GPCRs [5], we evaluated this functional process within the context of various D1R mutants that had specific alterations in their phosphorylation sites. Using a combination of truncations, forward (loss of phosphorylation) and reverse (gain of phosphorylation) mutations, along with two different β-arrestin recruitment assays, we determined that GRK-mediated phosphorylation of both residues comprising the T5/S6 cluster (T360 and S362) within the proximal C-terminus serves as a major driver for β-arrestin recruitment to the D1R. Parenthetically, it should be noted that the β-arrestin recruitment assays that we employed are BRET-based proximity assays that essentially quantify β-arrestin binding to the D1R. Notably, other GRK phosphorylation sites that were determined to be critically important for β-arrestin binding to the D1R include the S2/S3/S4 cluster in ICL3. These latter data confirm our previous fundings using a microcopy-based β-arrestin translocation assay [27]. In addition, our data suggest that other GRK-mediated phosphorylation sites within the middle (S7–9) or distal (T15/S16 and T17/S18/S19/T20) C-terminus can enhance β-arrestin binding to the D1R; however, phosphorylation of these sites alone is unable to create a high-affinity binding domain for β-arrestin. These data mostly agree with earlier reports previously showing that mutation of the C-terminus of the D1R (mostly truncations) impaired either receptor phosphorylation, desensitization, or internalization [23,25,26,27,28,29], although the specific phosphorylation sites were not (completely) identified, and the nature and role of ICL3 phosphorylation was not fully appreciated.

Having identified the GRK phosphorylation sites on the D1R that mediate β-arrestin interactions, we wished to investigate the role(s) of these sites in regulating receptor function. Several studies have illustrated the importance of β-arrestin in regulating D1R-mediated behaviors and downstream signaling pathways including the activation of ERK1/2 and Src [42,43,44,45,46,47]. Further, we have previously determined that β-arrestin is required for dopamine-stimulated D1R internalization [28]. Thus, as an initial approach, we examined the ability of various D1R phosphorylation mutants to undergo dopamine-stimulated endocytosis. Perhaps not surprisingly, we found that the GRK-null and Tail Total mutants, both of which lack GRK-mediated phosphorylation and β-arrestin recruitment, were severely impaired in their ability to undergo dopamine-induced internalization. What was surprising, however, were the results with the D1R T5V/S6A and D1R S2/S3/S4 mutants, both of which are partially impaired in their ability to recruit β-arrestin (cf. Figure 4c and Figure 6b). We found that the dopamine-stimulated internalization of the S2/S3/S4 mutant was comparably impaired as the GRK-null and Tail Total mutants whereas the T5V/S6A mutant internalized to almost the same extent as the WT D1R. These results imply that the ICL3 phosphorylation cluster plays two functional roles in β-arrestin-mediated receptor internalization. Firstly, this phosphorylated cluster contributes to the high-affinity binding of β-arrestin to the D1R. Secondly, the ICL3 cluster must be involved in producing an activated state of β-arrestin that triggers the endocytotic process. In contrast, the phosphorylated T5/S6 cluster appears to be primarily involved in the high-affinity binding of β-arrestin to the D1R and does not contribute to β-arrestin activation, at least as it pertains to its role in the endocytic process. Our observation that distinct D1R phosphorylation sites are involved in β-arrestin binding and activation agree well with current information on how GPCRs promote β-arrestin activation [4,48,49] and is analogous with the “bar-code” hypothesis of how different GRK phosphorylation sites can regulate GPCR-β-arrestin signaling [50,51].

Notably, Latorraca et al. (2020) have demonstrated that the number and spatial orientation of phosphorylated residues within a GPCR can modulate β-arrestin binding affinity and/or the ability to promote distinct active conformations of β-arrestin that can trigger various downstream signaling cascades [51]. In this regard, it is informative to compare our results with those of Kaya et al. (2020) [44]. These investigators did not examine D1R phosphorylation directly but rather examined the binding of β-arrestin to short peptides derived from the sequence of the human D1R. Phosphomimetic substitutions of either aspartate or glutamate residues were made with specific serines or threonines to mimic the negative charges of phosphate groups. Using this type of analysis, they identified a C-terminal peptide corresponding to D1R residues 426–440, where phosphomimetic substitution of residues corresponding to GRK sites 15–17 (shown in Figure 1) resulted in β-arrestin binding. They also identified an ICL3 peptide that included GRK sites S3 and S4 (Ser-258 and Ser-259) where phosphomimetic substitutions enhanced the binding of β-arrestin. Surprisingly, using intact cells, only mutations of the residues identified in the C-terminus affected β-arrestin recruitment to the full-length D1R; mutating residues in the ICL3 was without effect [44]. In contrast, when they examined D1R-mediated activation of ERK1/2 and Src, both of which are mediated by β-arrestin, the C-terminal mutations dampened both signaling pathways, whereas only the ICL3 mutations inhibited ERK1/2 activation. These data led to the conclusion that phosphorylation of the ICL3 promotes a conformation of β-arrestin that selectively targets the ERK1/2 signaling pathway. Similar results were recently found by Haider et al. (2022) who showed that different phosphorylation sites on the parathyroid hormone 1 receptor were involved in regulating ERK1/2 activation and receptor internalization, which are both β-arrestin-mediated processes [52].

The results of Kaya et al. (2020) [44] agree with our data showing that GRK phosphorylation of sites in the distal C-terminus can enhance β-arrestin binding to the D1R but, unfortunately, an analysis of GRK sites T5S6, which we believe to be major drivers of β-arrestin binding, was not performed. More interestingly, the observations by Kaya et al. (2020) [44] that ICL3 phosphorylation distinctly triggers activation of β-arrestin signaling through ERK1/2 mirror our observations concerning ICL3 phosphorylation directing β-arrestin-mediated receptor internalization. In support of the notion that ICL3 in the D1R can promote β-arrestin activation are biochemical data showing that this intracellular loop (as well as the C-terminus) can physically associate with β-arrestin [53]. Further, a recent cryo-EM structure of a neurotensin receptor 1 (NTSR1)-β-arrestin complex revealed interactions between phosphorylated residues in the C-terminal end of the ICL3 (similar in location to residues S2/S3/S4 in the D1R) and the finger loop region of β-arrestin that may be involved in the activation of β-arrestin [54].

In summary, we have mapped the GRK phosphorylation sites on the D1R and investigated how these sites relate to β-arrestin recruitment and binding to the receptor. We have further provided evidence that a cluster of phosphorylation sites in the ICL3 region are responsible for β-arrestin activation, leading to receptor internalization. Emerging structural data describing GPCR-β-arrestin interactions suggest that these results may be generalizable to multiple GPCRs.

## 4. Materials and Methods

### 4.1. Materials

[^3^H]SCH-23390 (86.00 Ci/mmol) and [^32^P]orthophosphate (carrier-free, 10 mCi/mL) were obtained from Perkin Elmer Life Sciences (Waltham, MA, USA). Cell culture media and reagents were purchased from Invitrogen (Carlsbad, CA, USA) and Corning (Glendale, AZ, USA). Calcium phosphate transfection kits were obtained from Clontech (Mountain View, CA, USA). MiniComplete^TM^ protease inhibitor cocktail was purchased from Roche Applied Science (Indianapolis, IN, USA). Receptor mutants were prepared either via QuikChange Multi Site-Directed Mutagenesis kit purchased from Stratagene (La Jolla, CA, USA) or at Bioinnovatise (Rockville, MD, USA). Mutagenesis primers were synthesized by Eurofins MWG Operon (Huntsville, AL, USA), and all mutations were verified by DNA sequencing. NuPage gels and buffers were purchased from Invitrogen. Cell culture flasks, materials, and all assay plates were purchased from ThermoFisher Scientific (Waltham, MA, USA) and Greiner Bio-One (Monroe, NC, USA). DNA constructs for the BRET assays were kind gifts from Dr. Jonathan A. Javitch. DA, anti-FLAG M2 affinity gel, and other fine chemicals and buffer components were purchased from Sigma-Aldrich (St. Louis, MO, USA), except where indicated.

### 4.2. Cell Culture and Transient Transfections

HEK293 cells were grown in Dulbecco’s modified Eagle’s medium (DMEM) supplemented with 10% fetal bovine serum and maintained at 37 °C in a humidified incubator containing 5% CO_2_. The day before transfection, cells were plated in 100 mm tissue culture dishes at 4 × 10^6^ cells/dish in serum-free DMEM. The cells were transfected with the indicated DNA constructs using polyethylenimine (PEI) as the transfection reagent at a ratio of 3:1 (μL PEI: μg DNA). Twenty-four hours after transfection, the medium was changed to DMEM supplemented with 10% fetal bovine serum, and cells were used for experiments the next day.

### 4.3. In Situ Phosphorylation Assays

In situ phosphorylation assays were performed as previously described [13,14,15]. Briefly, HEK293T cells were transfected with FLAG-tagged rat WT or mutant D1 receptors as described above. 24 h later, each transfection was reseeded into 2 wells of a 6-well Biocoat plate (BD Biosciences, San Jose, CA, USA). A portion of the transfection was retained in a 100 mm culture dish for radioligand binding assays as described below to determine the amount of receptor expressed in each treatment condition so that equal amounts of receptor could be loaded into each gel lane. The cells in the 6-well plate were incubated for 1 h in phosphate-free DMEM. The medium was then removed and replaced with 1 mL of fresh phosphate-free DMEM containing 106 μCi of [^32^P]H_3_PO_4_ for 1 h. Cells were then stimulated with either vehicle or 10 μM dopamine in the presence of 0.2 mM sodium metabisulfite for 15 min and then placed on ice. Cells were then washed twice with ice-cold EBSS and solubilized for in 1 mL of solubilization buffer (50 mM HEPES, 1 mM EDTA, 10% glycerol, 1% Triton X-100, pH 7.4, 50 mM NaF, 40 mM sodium pyrophosphate, and 150 mM NaCl) supplemented with Mini-Complete protease inhibitor cocktail for one hour at 4 °C. The samples were cleared by centrifugation and the protein concentration was determined using the BCA protein assay kit from Pierce. Equal amounts of receptor protein (generally 1–2 pmol of receptor) were resolved on 4 to 12% NuPage Bis-Tris gradient gels run in MOPS SDS running buffer (Invitrogen), dried, and subjected to autoradiography.

### 4.4. Radioligand Binding for In Situ Phosphorylation Assays

HEK293T cells were treated as described above, harvested and cell pellets were lysed in a dounce homogenizer in 5 mM Tris-HCl, pH 7.4 and 5 mM MgCl_2_, then centrifuged at 30,000× *g* for 30 min. Membranes were resuspended in 50 mM Tris-HCl, pH 7.4 and 100 µL of the suspension was added to tubes containing a receptor saturating concentration of [^3^H]SCH-23390 in a final volume of 1 mL. (+)-Butaclamol (3 µM) was used to determine nonspecific binding. Assay tubes were incubated at RT for 1.5 h then terminated by rapid filtration through GF/C filters pretreated with 0.3% polyethyleneimine. Bound radioactivity was quantified by liquid scintillation spectroscopy.

### 4.5. Radioligand Saturation Binding Assays

HEK293 cells that were transiently transfected with various WT and mutant D1R constructs, as described above, were removed mechanically using calcium-free Earle’s balanced salt solution (EBSS-). Intact cells were collected by centrifugation and then lysed with 5 mM Tris–HCl and 5 mM MgCl_2_ at pH 7.4. Homogenates were centrifuged at 30,000× *g* for 30 min. The membranes were resuspended in EBSS pH 7.4. For saturation binding studies, membrane preparations were incubated for 90 min at room temperature with various concentrations of [^3^H]-SCH-23390 in a reaction volume of 250 μL. Nonspecific binding was determined in the presence of 4 μM (+)-butaclamol. Bound ligand was separated from unbound by filtration through GF/C filters using a PerkinElmer, (Waltham, MA, USA), cell harvester and quantified using a TopCount instrument (PerkinElmer, Waltham, MA, USA). Appendix A shows the maximum (Bmax) binding and affinity (Kd) values for [^3^H]-SCH-23390 for the various D1R constructs used in this study.

### 4.6. CAAX β-Arrestin Recruitment BRET Assay

HEK293 cells were transiently transfected with the indicated D1R constructs, GFP-CAAX (which is anchored in the plasma membrane [35]), and β-arrestin2-Rluc2. Cells were harvested with EBSS-, plated in 96-well white plates at 20,000 cells/well in Dulbecco’s phosphate-buffered saline (DPBS) and incubated at room temperature for 45 min. Cells were then incubated with 2 µM Prolume Purple (Nanolight Technology, Pinetop, AZ, USA) for 5 min, then stimulated with the indicated concentrations of dopamine for 5 min. The BRET signal was determined by quantifying and calculating the ratio of the light emitted by GFP (515 nm) over Rluc2 (410 nm) using a PHERAstar FSX Microplate Reader (BMG Labtech, Cary, NC, USA).

### 4.7. Direct β-Arrestin Recruitment BRET Assay

HEK293 cells were transiently transfected with the indicated D1R-Rluc8 constructs and β-arrestin2-mVenus. Cells were harvested with EBSS-, plated in 96-well white plates at 20,000 cells/well in Dulbecco’s phosphate-buffered saline (DPBS) and incubated at room temperature for 45 min. Cells were incubated with 5 μM coelenterazine h (Nanolight Technology, Pinetop, AZ, USA) for 5 min and then stimulated with the indicated concentrations of dopamine for 5 min. The BRET signal was determined by quantifying and calculating the ratio of the light emitted by mVenus (525 nm) over that emitted by Rluc8 (485 nm) using a PHERAstar FSX Microplate Reader (BMG Labtech, Cary, NC, USA).

### 4.8. cAMP CAMYEL BRET Assay

HEK293 cells transiently expressing the indicated D1R constructs and the CAMYEL cAMP biosensor (yellow fluorescence protein-Epac-Rluc) [55] were harvested with EBSS, plated in 96-well white plates at 20,000 cells/well in DPBS and incubated at room temperature for 45 min. Cells were incubated with 5 μM coelenterazine h (Nanolight Technology, Pinetop, AZ, USA) for 5 min, followed by stimulation with the indicated concentrations of the dopamine for 5 min. The BRET signal was determined by quantifying and calculating the ratio of the light emitted by mVenus (525 nm) over that emitted by Rluc8 (485 nm) using a PHERAstar FSX Microplate Reader (BMG Labtech, Cary, NC, USA).

### 4.9. Lyn Kinase BRET Internalization Assay

HEK293 cells were transiently transfected with the indicated Rluc8-tagged D1R constructs and Lyn-rGFP ([35]). Cells were harvested with EBSS, resuspended in BRET buffer (DPBS), plated in white 96-well plates (Greiner Bio-One), and incubated at room temperature for 30 min. Cells were stimulated with indicated concentrations of dopamine for 25 min, followed by incubation with 2 μM Prolume Purple (NanoLight Technology, Pinetop, AZ, USA) for 5 min. The BRET signal was determined by quantifying and calculating the ratio of the light emitted by rGFP (515/30 nM) over that emitted by Rluc8 (410/80 nM) using a PHERAstar FSX Microplate Reader (BMG Labtech, Cary, NC, USA).

### 4.10. Statistical Analyses

Nonlinear regression analyses were conducted using GraphPad Prism version 9.3.1 (GraphPad Software, Inc., La Jolla, CA, USA). Results are expressed as the mean ± SEM values. EC50 and Emax values were calculated from individual concentration–response curves and then averaged to generate mean ± SEM values. For all multiple comparisons statistical tests, the reported *p* values are multiplicity-adjusted *p* values. Other statistical tests were performed as described in the legends using GraphPad Prism.

## Figures and Tables

**Figure 2 ijms-24-06599-f002:**
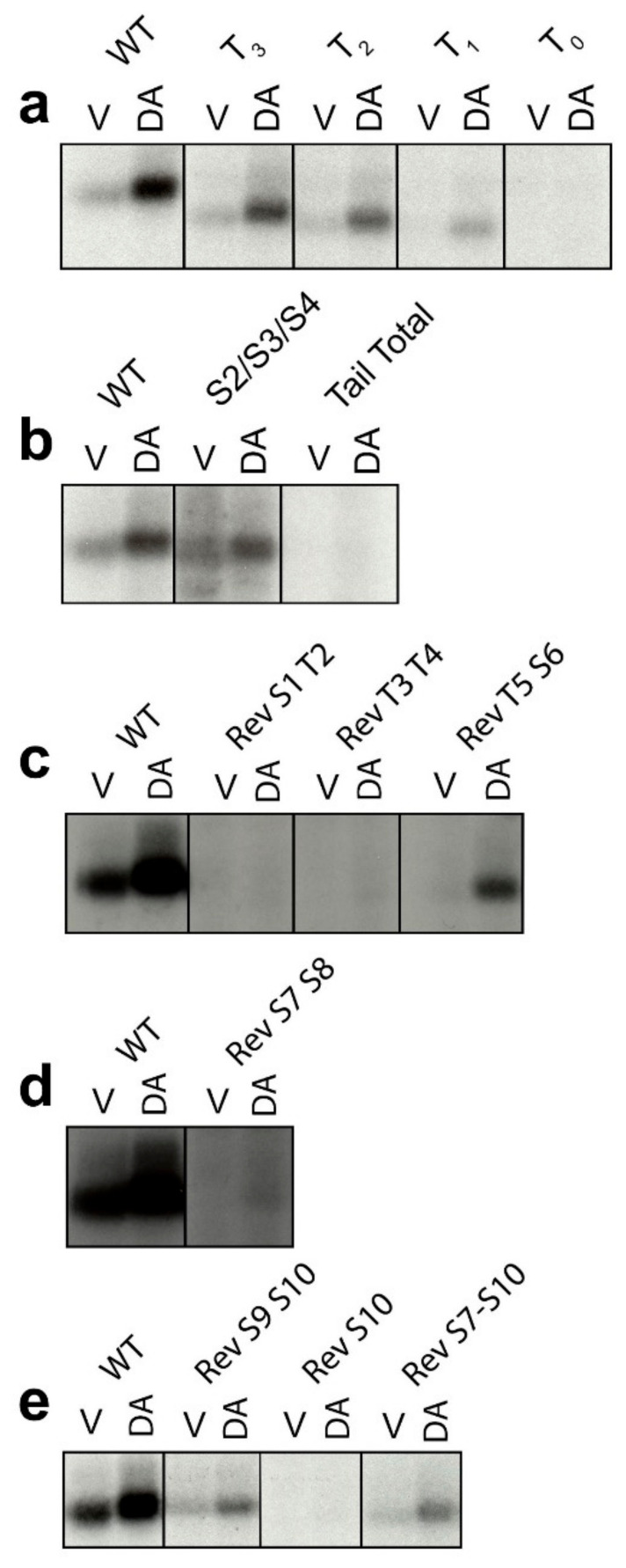
Mutational analyses reveal dopamine-stimulated (GRK-mediated) D1R phosphorylation sites. In situ phosphorylation assays were performed as described in Section 4. Briefly, wild-type (WT), site-specific, reverse (Rev), or C-terminal truncation mutants of the D1R were treated with either vehicle (labeled “V”) or 10 µM dopamine (labeled “DA”) for 15 min prior to cell lysis, immunoprecipitation, resolution by SDS-PAGE, and autoradiography. Within each panel, equal amounts of D1R protein, as quantified by radioligand binding, were loaded into each lane of the gels. Representative experiments are shown, which were performed at least twice with similar results. Numbering scheme for the site-specific, reverse (Rev), and truncation mutants are shown in the receptor diagram in Figure 1. (**a**) The D1R WT and C-terminal truncation mutants T_0_, T_1_, T_2_ and T_3_ are shown. (**b**) The D1R WT, triple ICL3 mutant S2/S3/S4, and mutant Tail Total (in which all 20 C-terminal serines and threonines are mutated) are shown. (**c**) The D1R WT and reverse mutants Rev S1T2, Rev T3T4, and Rev T5T6 are shown. (**d**) The D1R WT and reverse mutant Rev S7S8 are shown. (**e**) The D1R WT and reverse mutants Rev S9S10, Rev 10, and Rev S7–S10 are shown.

**Figure 3 ijms-24-06599-f003:**
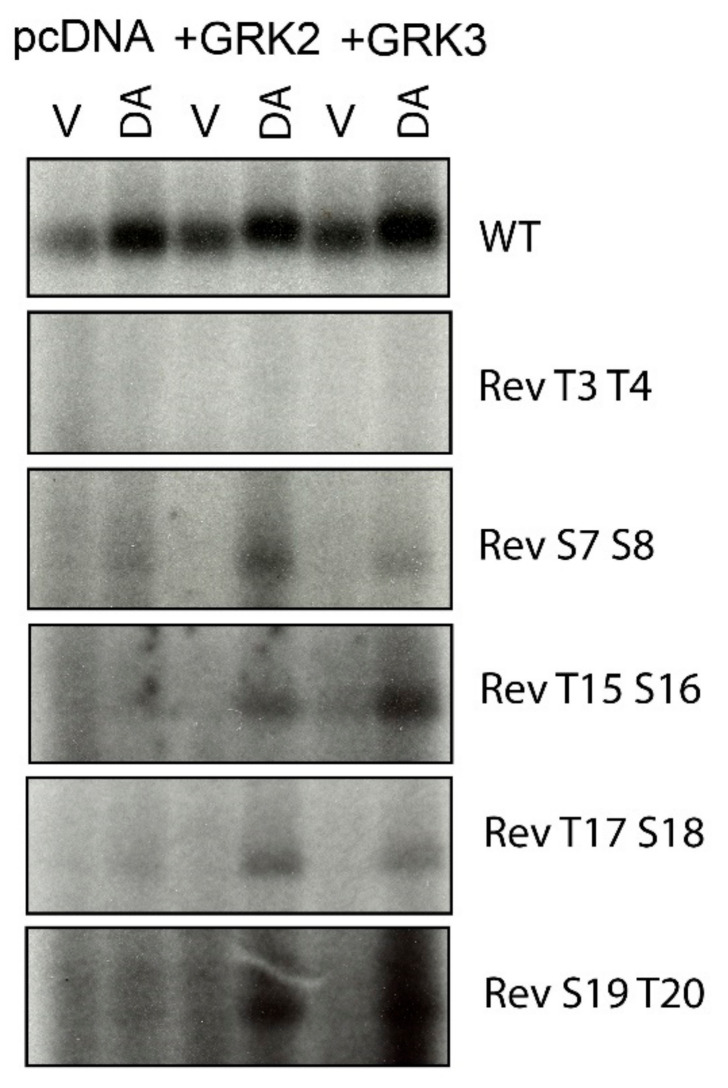
Overexpression of GRK2 or GRK3 can enhance dopamine-induced phosphorylation of specific D1R residues. In situ phosphorylation assays were performed as described in Section 4. Cells were transfected with either empty vector, GRK2, or GRK3 along with the D1R WT and site-specific reverse mutants of the C-terminus. Cells were treated with either vehicle (labeled “V”) or 10 µM dopamine (labeled “DA”) for 15 min prior to lysis, immunoprecipitation, resolution by SDS-PAGE, and autoradiography. Equal amounts of D1R protein, as quantified by radioligand binding, were loaded into each lane of the gel. Representative experiments are shown, which were performed at least twice with similar results. Numbering scheme for the site-specific reverse (Rev) mutants are shown in Figure 1.

**Figure 4 ijms-24-06599-f004:**
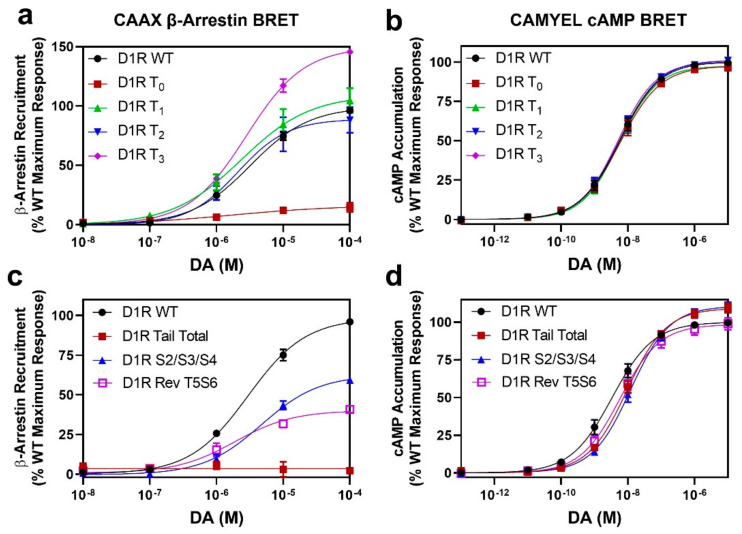
β-arrestin recruitment is impaired by specific D1R mutations while G protein signaling is unaffected. CAAX β-arrestin recruitment and CAMYEL cAMP accumulation assays were performed as described in Section 4. Data are expressed as a percentage of the maximum D1R WT DA response in each experiment and are displayed as the mean ± SEM of at least three experiments, performed in triplicate. Statistical comparisons between the following D1R WT and mutant curve parameters were made using a one-way ANOVA with Dunnett’s post hoc comparison: * *p* < 0.05, ** *p* < 0.01, *** *p* < 0.001, and **** *p* < 0.0001. (**a**) The four D1R truncation mutants are compared to the WT receptor for DA-induced β-arrestin recruitment yielding the following curve parameters: D1R WT EC_50_ = 3.3 ± 0.3 µM, Emax = 100%; D1R T_0_ EC_50_ = 2.8 ± 2.6 µM, Emax = 16 ± 4.4% ****; D1R T_1_ EC_50_ = 2.8 ± 0.7 µM, Emax = 110 ± 9.3%; D1R T_2_ EC_50_ = 2.4 ± 0.5 µM, Emax = 90 ± 10%; D1R T_3_ EC_50_ = 2.9 ± 0.4 µM, Emax = 150 ± 0.6% ***. (**b**) None of the C-terminal truncation mutants affect the ability of the receptor to signal via G proteins (cAMP accumulation): D1R WT EC_50_ = 6.1 ± 1.5 nM, Emax = 100%; D1R T_0_ EC_50_ = 6.5 ± 2.4 nM, Emax = 97 ± 2.3%; D1R T_1_ EC_50_ = 6.0 ± 1.6 nM, Emax = 97 ± 1.4%; D1R T_2_ EC_50_ = 6.2 ± 2.0 nM, Emax = 101 ± 1.8%; D1R T_3_ EC_50_ = 5.3 ± 1.3 nM, Emax = 100 ± 1.9%. (**c**) Mutating specific serine and/or threonine residues in the ICL3 (D1R S2/S3/S4) or C-terminus (D1R Tail Total or Rev T5S6) variably affects β-arrestin recruitment: D1R WT EC_50_ = 3.3 ± 0.5 µM, Emax = 100%; D1R Tail Total EC_50_ = undefined, Emax = no response ****; D1R S2/S3/S4 EC_50_ = 4.7 ± 0.9 µM, Emax = 61 ± 0.8% ****; D1R Rev T5S6 EC_50_ = 3.0 ± 0.2 µM, Emax = 46 ± 4.4% ****. (**d**) The D1R Tail Total, S2/S3/S4, and Rev T5S6 mutations have minimal effects on the G protein interactions: D1R WT EC_50_ = 4.3 ± 1.1 nM, Emax = 100 ± 0.1%; D1R Tail Total EC_50_ = 9.8 ± 1.7 nM, Emax = 109 ± 3.1% *; D1R S2/S3/S4 EC_50_ = 13 ± 2.8 nM **, Emax = 110 ± 2.5% **; D1R Rev T5S6 EC_50_ = 6.5 ± 2.1 nM, Emax = 98 ± 3.9%.

**Figure 5 ijms-24-06599-f005:**
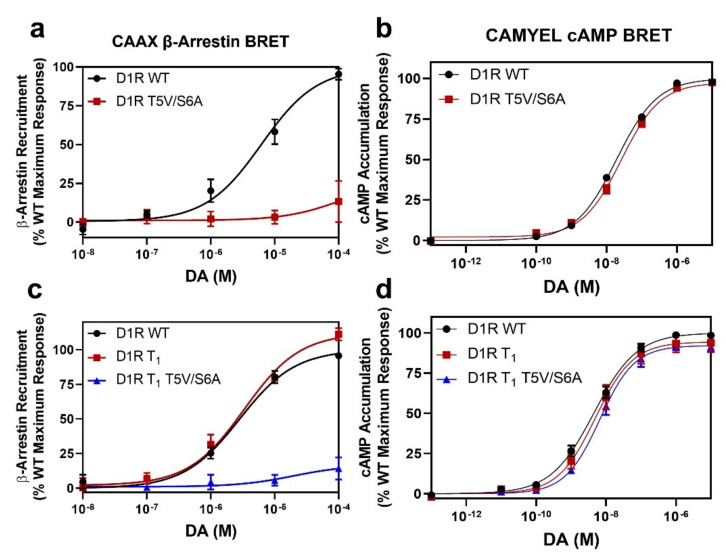
The T5S6 C-terminal phosphorylation sites are major drivers of β-arrestin recruitment to the D1R. CAAX β-arrestin recruitment and CAMYEL cAMP accumulation assays were performed as described in Section 4. Data are expressed as a percentage of the maximum D1R WT DA response in each experiment and are displayed as the mean ± SEM of at least three experiments, performed in triplicate. Statistical comparisons between the following D1R WT and mutant curve parameters were made using a one-way ANOVA with Dunnett’s post hoc comparison: * *p* < 0.05, *** *p* < 0.001, and **** *p* < 0.0001. (**a**) The T5V/S6A mutation of the full-length D1R also severely diminishes β-arrestin recruitment: D1R WT EC_50_ = 7.0 ± 2.5 µM, Emax = 100%; D1R T5V/S6A EC_50_ > 100 µM ****, Emax = 18 ± 8.8% ***. (**b**) The T5V/S6A mutation has no effect on the ability of the receptor to signal via G proteins: D1R WT EC_50_ = 20 ± 1.7 nM, Emax = 100%; D1R T5V/S6A EC_50_ = 27 ± 3.1 nM, Emax 98 ± 2.3%. (**c**) Mutation of both the T5 and S6 residues to valine or alanine, respectively, within the context of the D1R T_1_ C-terminal mutant, severely diminishes β-arrestin recruitment: D1R WT EC_50_ = 2.9 ± 0.2 µM, Emax = 100%; D1R T_1_ EC_50_ = 3.4 ± 0.8 µM, Emax = 112 ± 5.1%; D1R T_1_ + T5V/S6A EC_50_ > 100 µM *, Emax = 17 ± 10.5% ***. (**d**) The T5V/S6A mutations have no effect on the ability of the D1R T_1_ truncation mutant to signal via G proteins: D1R WT EC_50_ = 4.6 ± 0.8 nM, Emax = 100%; D1R T_1_ EC_50_ = 5.4 ± 1.3 nM, Emax = 94 ± 5.3%; D1R T_1_ + T5V/S6A EC_50_ = 7.1 ± 1.5 nM, Emax 92 ± 2.9%.

**Figure 6 ijms-24-06599-f006:**
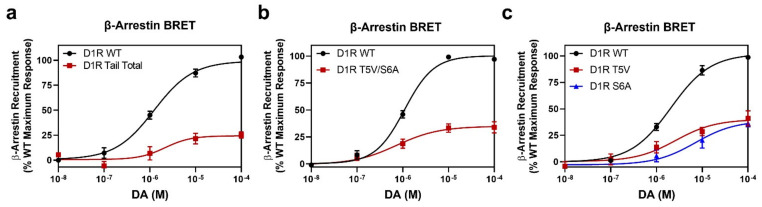
Orthogonal assays of β-arrestin recruitment confirm that mutation of all C-terminal phosphorylation sites, or T5 and/or S6, severely diminishes β-arrestin recruitment. Direct D1R-β-arrestin BRET assays were performed as described in Section 4. Data are expressed as a percentage of the maximum D1R WT response to DA stimulation in each experiment and are displayed as the mean ± SEM of at least three experiments, performed in triplicate. Statistical comparisons between the following D1R WT and mutant curve parameters were made using a one-way ANOVA with Dunnett’s post hoc comparison: ***** *p* < 0.0001. (**a**) In this direct β-arrestin recruitment assay, the D1R Tail Total mutant displays diminished DA-induced β-arrestin recruitment: D1R WT EC_50_ = 1.3 ± 0.3 µM, Emax = 100%; D1R Tail Total EC_50_ = 4.0 ± 2.5 µM, Emax = 25 ± 3.1% ****. (**b**) β-Arrestin recruitment is also diminished by simultaneous mutation of residues T5 and S6: D1R WT EC_50_ = 1.1 ± 0.1 µM, Emax = 100%; D1R T5V/S6A EC_50_ = 1.1 ± 0.6 µM, Emax 36 ± 4.1% ****. (**c**) Mutation of either T5 or S6 to valine or alanine, respectively, diminishes β-arrestin recruitment to the D1R: D1R WT EC_50_ = 1.7 ± 0.3 µM, Emax = 100%; D1R T5V EC_50_ = 2.8 ± 1.1 µM, Emax = 39 ± 5.4% ****; D1R S6A EC_50_ = 5.5 ± 1.7 µM, Emax = 39 ± 2.8% ****.

**Figure 7 ijms-24-06599-f007:**
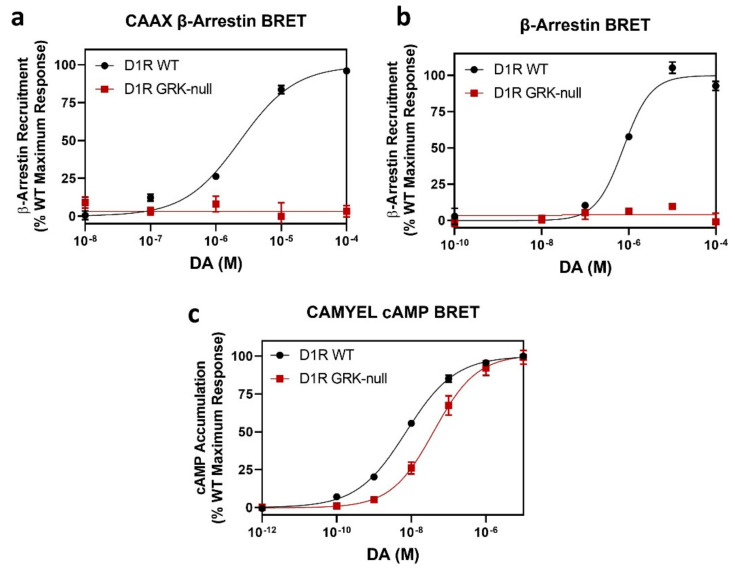
Creation of a “GRK-null” D1R through simultaneous mutation of the GRK-mediated phosphorylation sites ablates β-arrestin recruitment to the D1R. The GRK-null construct is delineated in Figure 1 with red lettering/numbering. CAAX β-arrestin recruitment, CAMYEL cAMP accumulation, and direct D1R-β-arrestin BRET assays were performed as described in Section 4. (**a**) CAAX β-arrestin recruitment assays: D1R WT EC_50_ = 2.4 ± 0.1 µM, Emax = 100%; D1R GRK-null EC_50_ = undefined, Emax = 4.4 ± 2.7% ****. (**b**) Direct D1R-β-arrestin recruitment BRET assays: D1R WT EC_50_ = 0.8 ± 0.07 µM, Emax = 100%; D1R GRK-null EC_50_ = undefined, Emax = 5.1 ± 3.0% ****. (**c**) CAMYEL cAMP accumulation assays: D1R WT EC_50_ = 7.4 ± 0.7 nM, Emax = 100%; D1R GRK-null EC_50_ = 45 ± 9.9 nM **, Emax = 101 ± 4.4%. Data are expressed as a percentage of the maximum D1R WT response to DA and are shown as the mean ± SEM of at least three experiments performed in triplicate. Statistical comparisons between the D1R WT and mutant curve parameters were made using a *t*-test: ** *p* < 0.01, and **** *p* < 0.0001.

**Figure 8 ijms-24-06599-f008:**
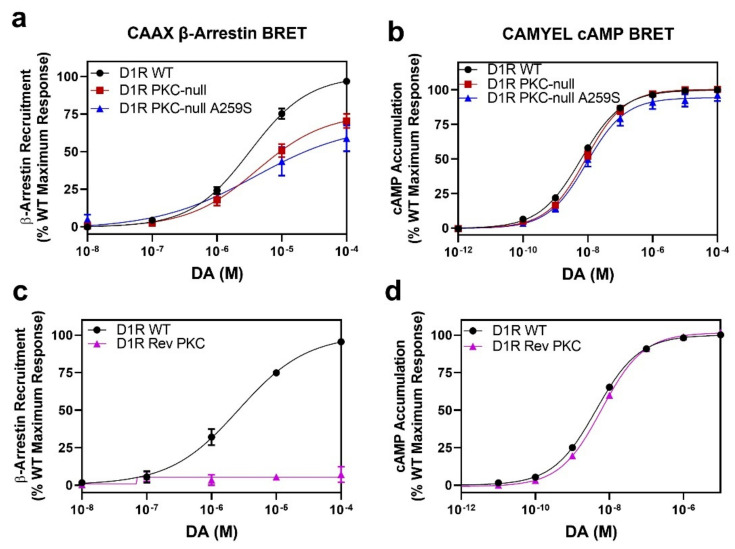
D1R residues phosphorylated by PKC have small but significant effects on DA-induced β-arrestin recruitment. CAAX β-arrestin recruitment and CAMYEL cAMP accumulation assays were performed as described in Section 4. (**a**) A “PKC-null” D1R in which residues S11–14 in the C-terminus and S4 in the ICL3 were simultaneously mutated to alanines displays a significant decrease in DA-induced β-arrestin recruitment: D1R WT EC_50_ = 3.4 ± 0.5 µM, Emax = 100%; D1R PKC-null EC_50_ = 4.5 ± 1.5 µM, Emax = 71 ± 3.5% ***. Reintroduction of serine S4 (S259) in the PKC-null mutant does not change DA-induced β-arrestin recruitment: D1R PKC-null + A259S EC_50_ = 3.2 ± 1.7 µM, Emax 59 ± 9.4% ****. (**b**) The PKC-null and PKC-null + A259 constructs display normal G protein signaling: D1R WT EC_50_ = 5.9 ± 0.7 nM, Emax = 100%; D1R PKC-null EC_50_ = 9.9 ± 1.5 nM, Emax = 101 ± 1.6%; D1R PKC-null A259S EC_50_ = 10 ± 1.6 nM, Emax 94 ± 4.4%. (**c**) A “reverse PKC” D1R construct in which residues S11, S12, S13, and S14 in the C-terminus were reverted to serines within the D1R Tail Total mutant does not display DA-induced β-arrestin recruitment: D1R WT EC_50_ = 2.6 ± 0.4 µM, Emax = 100%; D1R Rev PKC EC_50_ = undefined ****, Emax = no recruitment ****. (**d**) The D1R Rev PKC exhibits normal G protein interactions: D1R WT EC_50_ = 4.4 ± 0.3 nM, Emax = 100%; D1R Rev PKC EC_50_ = 6.2 ± 0.5 nM *, Emax = 102 ± 1.7%. Data are expressed as a percentage of the maximum WT response to DA and are shown as the mean ± SEM values of at least three experiments performed in triplicate. Statistical comparisons between the D1R WT and mutant curve parameters and between the D1R PKC-null and D1R PKC-null + A259S curve parameters in (**a**,**b**) were made using a one-way ANOVA with Sidak’s multiple post hoc comparison: WT vs. each mutant *** *p* < 0.0005, and **** *p* < 0.0001; D1R PKC-null vs. D1R PKC-null + A259S *p* > 0.05 in both (**a**,**b**). Statistical comparisons between WT and D1R Rev PKC curve parameters in (**c**,**d**) were made using a *t*-test: * *p* < 0.05 and **** *p* < 0.0001.

**Figure 9 ijms-24-06599-f009:**
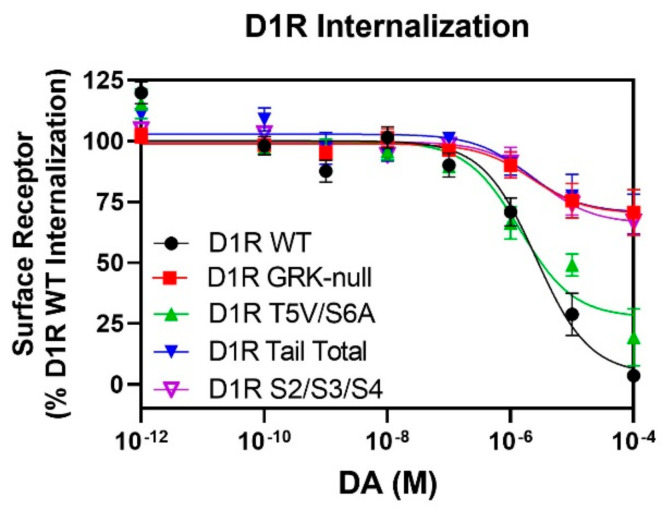
BRET assays of dopamine-induced D1R internalization indicate that the S2/S3/S4 cluster of serine residues in the ICL3 is necessary for efficient receptor internalization. Lyn BRET internalization assays were performed as described in Section 4. Data are expressed as a percentage of the maximum D1R WT response to DA in each experiment and are displayed as the mean ± SEM values of at least three experiments, performed in triplicate. Statistical comparisons between the following D1R WT and mutant curve parameters were made using a one-way ANOVA with Dunnett’s post hoc comparison: * *p* < 0.05 and ** *p* < 0.01. The Tail Total, GRK-null, and S2/S3/S4 mutants display severely impaired DA-induced receptor internalization, while the T5V/S6A mutant internalizes nearly to the same degree as WT: D1R WT EC_50_ = 5.1 ± 1.6 µM, Emax = 102 ± 1.9%; D1R GRK-null EC_50_ = undefined, Emax = 29 ± 10% *; D1R T5V/S6A EC_50_ = 2.4 ± 0.6 µM, Emax = 76 ± 9.6%; D1R Tail Total EC_50_ = undefined, Emax = 24 ± 10% *. D1R S2/S3/S4 EC_50_ = undefined, Emax = 34 ± 4.0% **.

## Data Availability

The data presented in this study are available on request from the corresponding author.

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
