# Peer review of "Delineation of G Protein-Coupled Receptor Kinase Phosphorylation Sites within the D1 Dopamine Receptor and Their Roles in Modulating β-Arrestin Binding and Activation"

_ijms, 2023, doi:10.3390/ijms24076599_

Round 1

Reviewer 1 Report

The authors examined the agonist-stimulated, GRK-mediated phosphorylation sites on the D1R by using detailed site-directed mutational analyses. The authors identified multiple GRK-mediated phosphorylated Ser and Thr residues on the C-terminus and ICL3 where the phosphorylation occurs in a hierarchical fashion for β-arrestin binding and subsequent β-arrestin-mediated receptor internalization, respectively. The experimental design looked very well and the manuscript is well written. However, there are two major  points (the 7th and 8th ones) and some ambiguous minor points. Please correct and address the following points.

1)    In L.67, “Thr-268, in the ICL3” is correct? In Figure 1, Thr-268 looks located near the intracellular end of TM6.

2)    In L.77–78, the authors described “constitutive phosphorylation by GRK4α takes place on five serine and threonine residues in the distal C-terminus of the D1R (Figure 1)”. However, there are six Ser/Thr residues in the distal C-terminus. Please show or describe which five residues (Thr-428, Ser-431, ,,) are targets for constitutive phosphorylation.

3)    In L.122, please describe Ser/Thr -> Ala/Val(?) mutation in the Tail Total mutant.

4)    In L.157–165, if possible, please add the data of the Rev S9 mutant for a clearer conclusion that S9 is the sole GRK phosphorylation site in the Rev S9S10 construct.

5)    In L.229, the authors described “different results (between GFP-CAAX+D1R β-arrestin2-Rluc2 recruitment assay and D1R-Rluc8 β-arrestin2-mVenus recruitment assay)”. Please clearly describe “different results (not statistically significant, P = ?? or which of statistical testing, P < ??)”. Similar resonance energy transfer rates between two different donor–acceptor pairs are expected? 

6)    In Figure 4b, d, please show dose-response curves normalized by the respective maximum responses for better readability of different EC50 values in Figures. In the legend, corresponding “Emax = XX ± YY%” would be “net Emax = XX ± YY% of the WT Emax” or another appropriate description. Please similarly correct Figures 5b, d, 7c, 8b, d, S2b, S3, and S4.

7)    In Materials and Methods, please explain the reason why some of the EC50 values of DA for D1R WT in CAMYEL cAMP BRET assays differed by ca. 2–5-fold (6.1 nM and 4.3 nM in Figure 4b and 4d, 20 nM and 4.6 nM in Figure 5b and 5d, 7.4 nM in Figure 7c, 5.9 nM and 4.4 nM in Figure 8b and 8d, 6.1 nM in Figure S2b, 8.8 nM, 4.8 nM, and 3.9 nM in Figure S3, 4.8 nM and 8.8 nM in Figure S4. Theoretically, the EC50 value of a given GPCR agonist is constant in the dose-response curves normalized by the maximum responses, although the interaction specificity of the GPCR mutant toward a single type of G protein coupling to a type of adenylate cyclase and resultant maximum cAMP accumulation levels would change. Much less expression of D1R or other critical experimental conditions such as various luminescence resonance energy transfer rates would alter the constant EC50 values. Please clearly explain this point.  

8)    In addition, please explain about the criterion for data exclusion due to abnormal expression ratio (this does not mean the emitted light ratio) of Rluc8 (525 nm) to mVenus (485 nM) or that of Rluc (535/30 nm) to YFP (475/30 nm) as described in the reference 55 (L.735 seemed wrong). There is inter-dish variability in expressed protein amounts and their relative ratios, which could alter S/N ratio and dopamine sensitivities of cAMP accumulation. Did the authors exclude culture dish with too weak bioluminescence condition or other experimental artifactual conditions?

9)    L.744 would be “of the light emitted by rGFP (515/30 nM) over that emitted by Rluc8 (410/80 nM) using a”?

10) Considering the minimized inter-transfected cell variability of donor/acceptor expression ratios, DA EC50 values of mutants for the cAMP BRET assay seemed significantly elevated by ca. 1.5–7-fold in D1R S234, T5V/S6A, GRK-null, Rev-S1T1, Rev-T3T4, Rev-S19T20 mutants. These results indicate that dopamine binding site and/or G-protein binding sites of these D1R mutants changed or the luminescence ratio calculation method was something wrong. Please clearly explain these points. In addition, please correct related parts such as L.304 and L.315 by adding statistical significance.

11) In L.294–301, I am curious about a possibility of rescue of DA-induced β-arrestin recruitment in the Rev T5S6S9, S1T2T5S6S9, and pS1pT2T5S6S9 (for a possible role of phosphorylated S1 and/or T2 in D1R desensitization and β-arrestin binding) mutants, respectively. If possible, please add these data in this or future report, although the data for the β-arrestin recruitment impairment by D1R T1 T5V/S6A, D1R T5V/S6A, D1R T5V and D1R S6A mutants are sufficient for the present conclusions.

12) In Figure 9, if possible, please add the data for D1R S234+GRK-null mutant for confirming a synergistic effect of the two hierarchically phosphorylated domains on D1R internalization. Without this data, the present conclusions are well supported by the present data.

Author Response

Response to Reviewer 1:

1)    In L.67, “Thr-268, in the ICL3” is correct? In Figure 1, Thr-268 looks located near the intracellular end of TM6.

The location of the juncture between ICL3 and TM6 with respect to the phospholipid membrane is not known with certainty, although the fact that Thr-268 serves as a phosphorylation site for PKA would argue that it is exposed to the cytoplasm. For convenience, we have used the “snake” diagram of the D1 receptor provided by GPCRdb.org.  However, we’re not sure that the TM regions it shows, which are predicted by hydrophobicity and secondary structure algorithms, are entirely accurate since a perfect alpha helix segment in a transmembrane protein is 24 residues in length and the GPCRdb.org diagram shows a 38 residue-long TM6. The length of a transmembrane segment can be affected by proline-induced kinks, however, and there are 2 prolines in TM6, so this is ambiguous. In recognition of this ambiguity, we have revised the relevant sentence as follows:

“Other second messenger-activated kinases that are known to phosphorylate the D1R include the cAMP-dependent protein kinase (PKA), which phosphorylates a single residue, Thr-268.”  in the ICL3.

2)    In L.77–78, the authors described “constitutive phosphorylation by GRK4α takes place on five serine and threonine residues in the distal C-terminus of the D1R (Figure 1)”. However, there are six Ser/Thr residues in the distal C-terminus. Please show or describe which five residues (Thr-428, Ser-431, ,,) are targets for constitutive phosphorylation.

 Thank you for pointing this out.  We now state on lines 77-80:

“Our laboratory has found that constitutive phosphorylation by GRK4a takes place on five serine and threonine residues (Thr-428, Ser-431, Ser-441, S445, and Thr-446) in the distal C-terminus of the D1R (Figure 1) and is associated with receptor desensitization and internalization [28].”

And in the legend to Figure 1 on lines 141-142 we now state:

“The sites that are constitutively phosphorylated by GRK4a are: T15, S16, S18, S19 and T20 [28].”

It should be noted that T17 was identified as a DA-stimulated GRK phosphorylation site in the current study.

3)    In L.122, please describe Ser/Thr -> Ala/Val(?) mutation in the Tail Total mutant.

The Tail Total mutant was described in two places in the manuscript. In the legend to Figure 1, there is the statement:

“The twenty serine and threonine residues in the C-terminus are numbered 1-20. Simultaneous mutation of these 20 residues refers to the “Tail Total” mutant.”

And in lines 114-119 of the text:

“Since the D1R C-terminus appears to be heavily phosphorylated, a reverse mutational approach was next used in which all 20 potential phosphorylation sites in this region were mutated from serines and threonines to alanines and valines, respectively (Figure 1). This mutant construct, referred to as “Tail Total”, provides a null background in which, notably, there is no DA-induced phosphorylation of the D1R (Figure 2b).”

4)    In L.157–165, if possible, please add the data of the Rev S9 mutant for a clearer conclusion that S9 is the sole GRK phosphorylation site in the Rev S9S10 construct.

This would be an interesting experiment to do, although it would significantly delay the publication of this manuscript. However, the observation in Figure 2e that there is no detectable D1R phosphorylation in the Rev S10 mutant, but significant phosphorylation in the Rev S9S10 mutant strongly suggests that it is S9, and not S10, that accounts for the observed phosphorylation.

5)    In L.229, the authors described “different results (between GFP-CAAX+D1R β-arrestin2-Rluc2 recruitment assay and D1R-Rluc8 β-arrestin2-mVenus recruitment assay)”. Please clearly describe “different results (not statistically significant, P = ?? or which of statistical testing, P < ??)”. Similar resonance energy transfer rates between two different donor–acceptor pairs are expected? 

We think this comment was due to a misunderstanding. We don’t think there is a difference in the results between the GFP-CAAX+D1R β-arrestin2-Rluc2 recruitment assay (Emax = 16 ± 4.4% vs. WT D1R) and the D1R-Rluc8 β-arrestin2-mVenus recruitment assay (Emax = 13 ± 6.5% vs. WT) using the D1R T0 truncation mutant. We were referring to a previous study in which the D1R T0  mutant was found to promote significant b-arrestin-GFP translocation to the plasma membrane using a microscopy-based assay. We feel, however, that the two different BRET assays used in the current study more accurately reflect the ability of the T0 mutant to recruit b-arrestin (it is severely impaired). To clarify, we now state on lines 228-233

“Since we previously found using live-cell microscopy [27], that this mutant construct, when stimulated with DA, promoted subcellular translocation of GFP-tagged β-arrestin, we turned to an orthogonal β-arrestin recruitment BRET assay (Figure S1), and, similar to the CAAX β-arrestin assay results, we found that the D1R T0 mutant displays severely impaired DA-induced β-arrestin recruitment (Emax = 13 ± 6.5% vs. WT, p = 0.0002).”

And on lines 233-238, we speculated on why the microscopy-based assay may have provided different results:

“We are not certain why different results were observed using the microscopy-based assay [27], but it might involve heterogeneous expression of the assay plasmids and difficulty in quantifying single cell data using microscopy. In contrast, the BRET assays used in this study quantify the results from the entire population of cells in the assay plate and are more reliable and quantitative.”

6)    In Figure 4b, d, please show dose-response curves normalized by the respective maximum responses for better readability of different EC50 values in Figures. In the legend, corresponding “Emax = XX ± YY%” would be “net Emax = XX ± YY% of the WT Emax” or another appropriate description. Please similarly correct Figures 5b, d, 7c, 8b, d, S2b, S3, and S4.

In each of the figures noted, we have already expressed the data as suggested by the reviewer. In each experiment where we investigated the dopamine-stimulated response of a mutant receptor, we ran a WT control in parallel and expressed all data as a percentage of the maximum response to dopamine using the WT receptor. This is readily apparent in cases that involved a change in Emax values such as in Figures 4a and 4c  Notably, the following phrase was included in all of the figure legends (except for Figure S2, which is now corrected):

“Data are expressed as a percentage of the maximum D1R WT DA response in each experiment and are displayed as mean ± SEM of at least three experiments, performed in triplicate.”

7)    In Materials and Methods, please explain the reason why some of the EC50 values of DA for D1R WT in CAMYEL cAMP BRET assays differed by ca. 2–5-fold (6.1 nM and 4.3 nM in Figure 4b and 4d, 20 nM and 4.6 nM in Figure 5b and 5d, 7.4 nM in Figure 7c, 5.9 nM and 4.4 nM in Figure 8b and 8d, 6.1 nM in Figure S2b, 8.8 nM, 4.8 nM, and 3.9 nM in Figure S3, 4.8 nM and 8.8 nM in Figure S4. Theoretically, the EC50 value of a given GPCR agonist is constant in the dose-response curves normalized by the maximum responses, although the interaction specificity of the GPCR mutant toward a single type of G protein coupling to a type of adenylate cyclase and resultant maximum cAMP accumulation levels would change. Much less expression of D1R or other critical experimental conditions such as various luminescence resonance energy transfer rates would alter the constant EC50 values. Please clearly explain this point.

The important thing to note is that the G protein-mediated signaling assays involved transient transfection experiments performed over a multi-year period. As such, there can be drift in the cellular expression levels of various signaling components involved in increasing the intracellular levels of cAMP. While we always have a handle on the receptor expression levels using radioligand binding assays, we don’t have a handle on the expression levels of G proteins or adenylyl cyclases (or other modulators such as GRKs, RGSs, etc.) that might alter the cAMP response. In a highly amplified receptor signaling system, such as the CAMYEL cAMP assay, it would not be surprising to see small alterations (2-5-fold) in agonist EC50 values between such experiments, especially over time. What is important, however, is that, as described above, in each experiment involving a mutant receptor(s), we always ran a WT control and expressed our data relative to the WT values and, where appropriate, performed statistical analyses to determine if there was an effect of the mutation. As such, we believe that the controls and our results are valid.

8)    In addition, please explain about the criterion for data exclusion due to abnormal expression ratio (this does not mean the emitted light ratio) of Rluc8 (525 nm) to mVenus (485 nM) or that of Rluc (535/30 nm) to YFP (475/30 nm) as described in the reference 55 (L.735 seemed wrong). There is inter-dish variability in expressed protein amounts and their relative ratios, which could alter S/N ratio and dopamine sensitivities of cAMP accumulation. Did the authors exclude culture dish with too weak bioluminescence condition or other experimental artifactual conditions?

We do not routinely exclude any culture dishes so have not established criteria for doing so. Reference 55 was provided solely to note the source of the CAMYEL plasmid.

9)    L.744 would be “of the light emitted by rGFP (515/30 nM) over that emitted by Rluc8 (410/80 nM) using a”?

This correction has been made.  The sentence now reads (L.756):

“The BRET signal was determined by quantifying and calculating the ratio of the light emitted by rGFP (515/30 nM) over that emitted by Rluc8 (410/80 nM) using a PHERAstar FSX Microplate Reader (BMG Labtech, Cary, NC).”

10) Considering the minimized inter-transfected cell variability of donor/acceptor expression ratios, DA EC50 values of mutants for the cAMP BRET assay seemed significantly elevated by ca. 1.5–7-fold in D1R S234, T5V/S6A, GRK-null, Rev-S1T1, Rev-T3T4, Rev-S19T20 mutants. These results indicate that dopamine binding site and/or G-protein binding sites of these D1R mutants changed or the luminescence ratio calculation method was something wrong. Please clearly explain these points. In addition, please correct related parts such as L.304 and L.315 by adding statistical significance.

There were no statistically significant effects on the DA curve parameters in the cAMP assay for the D1R T5V/S6, Rev S1T2, Rev T3T4, or Rev S19T20 mutants, as questioned. There were minor, but statistically significant, effects on the DA curve parameters in the cAMP assay for the D1R Tail Total, S234, GRK-null, Rev S7-S10, and Rev S17S18 mutants. These statistically significant differences were indicated in the figure legends (and now in the text), although the effects were so small that we didn’t consider them to be biologically significant.  We now state in the text on lines 304-307-:

“These D1R mutants exhibited normal G protein interactions (Figure S3 and S4) except for the Rev S7-S10 and Rev S17S18 mutants, which each exhibited a ~2-fold (p < 0.05) increase in the DA EC50 value (Figures S3 and S4).”

We further state on lines 311-315:

“Unlike the effects of these mutations on β-arrestin recruitment, they all displayed little to no effect on G-protein signaling, although the Tail Total and S2/S3/S4 mutants exhibited statistically significant increases (~10%) in the DA Emax values (p < 0.05) and the DA EC50 value for the S2/S3/S4 mutant was increased by ~3-fold (p < 0.05) (Figure 4D).

With respect to the GRK-null cAMP results, which involved a significant 6-fold shift in the DA EC50 value, we have now expanded our statement on lines 403-404:

“We did notice, however, that this mutant exhibited a slightly reduced potency for DA in the cAMP accumulation assay (Figure 7c), which may be due to its reduced expression level (Table S1) or due to a direct effect on receptor-G protein interactions”

In the amplified cAMP assay (meaning that there are spare receptors), a reduction in receptor expression will result in a right-ward shift in the dose-response curve until receptor reserve is exhausted and we feel that this is the most likely explanation.

11) In L.294–301, I am curious about a possibility of rescue of DA-induced β-arrestin recruitment in the Rev T5S6S9, S1T2T5S6S9, and pS1pT2T5S6S9 (for a possible role of phosphorylated S1 and/or T2 in D1R desensitization and β-arrestin binding) mutants, respectively. If possible, please add these data in this or future report, although the data for the β-arrestin recruitment impairment by D1R T1 T5V/S6A, D1R T5V/S6A, D1R T5V and D1R S6A mutants are sufficient for the present conclusions.

The experiments suggested by the reviewer are interesting and might be pursued in a future study.

12) In Figure 9, if possible, please add the data for D1R S234+GRK-null mutant for confirming a synergistic effect of the two hierarchically phosphorylated domains on D1R internalization. Without this data, the present conclusions are well supported by the present data.

Actually, the GRK-null mutant already includes the S2A/S3A/S4A mutations.  This is stated in the legend to Figure 1 (lines 137-141):

“Simultaneous mutation of just the three Ser residues in the ICL3 refers to the “S2/S3/S4” mutant. The GRK-mediated D1R phosphorylation sites determined in this study, as well as in Rankin et al. (2006) [28] are delineated using red numbering and lettering on the receptor diagram. Simultaneous mutation of these residues refers to the “GRK-null” mutant.”

And in lines 393-398 of the text: (now revised))

“Based on our analyses above, we have identified D1R residues T5, S6, S7, S8, S9, T15, S16, T17, S18, S19, and T20 in the C-terminus, and residues S2, S3, and S4 in the ICL3 as being phosphorylated by GRKs upon stimulation with DA. We thus created a “GRK-null” D1R mutant in which these residues were simultaneously mutated to either alanines or valines and investigated the aggregate effects of these mutations on D1R function.”

Reviewer 2 Report

Moritz et al. investigated the G protein-coupled receptor kinase (GRK) phosphorylation sites of D1 dopamine receptor (D1R) via these sites’ relationships with beta-arrestin. By conducting mutational and in situ phosphorylation analyses, they demonstrated that there are 14 GRK-mediated phosphorylation sites of D1R, including serine and threonine residues inside the C-terminus and the 3rd intracellular loop (ICL3) of the receptor. They also revealed that the occurrence of phosphorylation on the C-terminus precedes that of the ICL3. Further beta-arrestin recruitment assays identified a cluster of phosphorylation sites close to the C-terminus, which are involved in the beta-arrestin binding with D1R. Their further experiments supported that the ICL3 phosphorylation sites are important for beta-arrestin activation and receptor internalization. In summary, this paper uses classical molecular experiments uncovered novel phosphorylation sites of D1R that have play key roles in the interaction between D1R and beta-arrestin. I only have some minor suggestions for improvement of the manuscript.

(1)   In the introduction, the authors provided detailed published research about GRK-mediated phosphorylation sites within D1R. However, this may be too scientific, thus lacking generalizability to common readers. For example, the ‘constitutive in nature’ would be replace with lay language that show enough information instead of just citing previous publications. Please consider simplifying the instruction and put some of these information into the discussion section.

(2)   The severe truncation, T0, is very important, and thus it is necessary to describe more about how and which sites are deleted in the protein coding sequences. In Figure 1, please annotate T0 in more details and point out the residue S4 with its full name such as S4 (S259) in the diagram of D1R.

(3)   At line 118, the labels for three mutation sites, including S256, S258, and S259, are represented by S2, 3, and 4 in the figure 1. However, the authors use ‘S234’ to represent them in the main text while labeling them as ‘S2’, ‘S3’, and ‘S4’. It is better to keep constant between figure legend and main text. The main purpose of these minor revisions would enhance the readability of the paper.

(4)   For figure 2, It would be more intuitive to demonstrate each mutate construct in carton along with the phosphorylation results. The same would be applicable to Figure 3. By visualization of each construct or additional treatment to it would make readers easier to digest the experimental results.

(5)   Did the author perform rescue experiment, such as using site directed mutagenesis to repair the mutation (T0, T5V/S6A, and other key sites affecting beta-arrestin recruitment) back to original coding sequence and treating the newly rescued group as a control? I strong suggest the author consider this, which is a standard to support the functions of these identified phosphorylation sites that are involved in beta-arrestin recruitment. If it is not possible, the author may need to discuss this as limitations in the discussion section.

(6)   In Supplementary Figure S2b, it seems that D1R WT and D1R S13-20 are similar in terms of cAMP Accumulation, because the two curves in the figures are very close to each other. How can the authors obtain the statistical Emax results. It would be necessary to add more information in the legend to explain it. For example, the author can add how Emax was calculated based on the data available in the figures.

(7)   For Figure 4, please use multiple adjusted p values to indicate significance.

(8)   In Figure S3 and S4, please use multiple adjustment p values to determine significance. Additionally, more information about these reverse mutants compared to the WT can be included as notes in the legend for better understanding and comparisons.

(9)   At lines 237-255, please add the statistical significance p value for each comparison. For other comparisons in the main test, please include significance p values accordingly.

Author Response

Response to Reviewer 2:

(1)   In the introduction, the authors provided detailed published research about GRK-mediated phosphorylation sites within D1R. However, this may be too scientific, thus lacking generalizability to common readers. For example, the ‘constitutive in nature’ would be replace with lay language that show enough information instead of just citing previous publications. Please consider simplifying the instruction and put some of these information into the discussion section.

We understand that the Reviewer found our Introduction to be quite detailed in nature, but it is critical that we review the historical, yet incomplete, work on D1 receptor phosphorylation in order to set the stage for our current investigation. Doing this in the Introduction provides the requisite context to our study in way that would not be achieved through adding this information to the Discussion. We are thus opting not to significantly alter the Introduction as requested, however, we have now defined the term “constitutive in nature” as it relates to our study.  On lines 72-74, we now state:

“Whereas GRK2, GRK3, and GRK5 were found to phosphorylate the D1R in an agonist-dependent fashion [23,26-28], D1R phosphorylation by GRK4 (specifically the GRK4a isoform) was found to be independent of agonist stimulation, or constitutive in nature [28].”

(2)   The severe truncation, T0, is very important, and thus it is necessary to describe more about how and which sites are deleted in the protein coding sequences. In Figure 1, please annotate T0 in more details and point out the residue S4 with its full name such as S4 (S259) in the diagram of D1R.

In the legend to Figure 1 on lines 132-134, we had previously stated and now further write:

“Four carboxyl terminal truncation mutants were also generated by inserting stop codons after amino acids 347 (T0), 370 (T1), 394 (T2), or 404 (T3). The T0 truncation site was based on the observation that Cys-346 in the rat D1R is palmitoylated.

Also, the amino acid numbers for S2, S3, and S4 are now given on line 137-138 in the Figure 1 legend:

“Simultaneous mutation of just the three Ser residues (S256, S258, and S259) in the ICL3 refers to the “S2/S3/S4” mutant.”

(3)   At line 118, the labels for three mutation sites, including S256, S258, and S259, are represented by S2, 3, and 4 in the figure 1. However, the authors use ‘S234’ to represent them in the main text while labeling them as ‘S2’, ‘S3’, and ‘S4’. It is better to keep constant between figure legend and main text. The main purpose of these minor revisions would enhance the readability of the paper.

As noted above, we have now changed the designation of the S234 mutant to S2/S3/S4/ throughout the text and figures as suggested by the reviewer.

(4)   For figure 2, It would be more intuitive to demonstrate each mutate construct in carton along with the phosphorylation results. The same would be applicable to Figure 3. By visualization of each construct or additional treatment to it would make readers easier to digest the experimental results.

We appreciate the reviewer’s comment, although there is no room in Figures 2 and 3 to put segments of the receptor due to the number of mutants in each panel/figure. This was why we created a detailed Figure 1 schematic showing the receptor mutations and labeling the individual phosphorylation sites for easy reference.

(5)   Did the author perform rescue experiment, such as using site directed mutagenesis to repair the mutation (T0, T5V/S6A, and other key sites affecting beta-arrestin recruitment) back to original coding sequence and treating the newly rescued group as a control? I strong suggest the author consider this, which is a standard to support the functions of these identified phosphorylation sites that are involved in beta-arrestin recruitment. If it is not possible, the author may need to discuss this as limitations in the discussion section.

Repairing the T0 mutation would simply recreate the WT receptor. As for the individual site mutations, we did indeed perform rescue experiments (we refer to these as reverse (Rev) mutations) to ascertain their effects on both phosphorylation and receptor function. These data are shown in Figures 2-4, 8, S3, and S4.  The data for reverse mutant Rev T5/S6 are shown in Figure 4 c/d and discussed on lines 308-312.

(6)   In Supplementary Figure S2b, it seems that D1R WT and D1R S13-20 are similar in terms of cAMP Accumulation, because the two curves in the figures are very close to each other. How can the authors obtain the statistical Emax results. It would be necessary to add more information in the legend to explain it. For example, the author can add how Emax was calculated based on the data available in the figures.

We have now added the following statement to the legend for Figure S2 that explains how the data were calculated/expressed:

“Data are expressed as a percentage of the maximum D1R WT DA response in each experiment and are displayed as mean ± SEM of at least three experiments, performed in triplicate.”

The Emax value for the D1R S13-20 mutant significantly differed from the WT control (Emax = 103 ± 0.6% vs. WT, p < 0.05) because the variance between the experiments was quite low resulting in a statistically significant result. While we report this statistical result in the figure legend, we do not believe that this small effect is biologically meaningful.

(7)   For Figure 4, please use multiple adjusted p values to indicate significance.

All of the statistics that we performed using multiple comparison methods provided p values that were adjusted. We now state in the Methods section on lines 762-763.

For all multiple comparisons statistical tests, the reported p values are multiplicity adjusted p values. 

(8)   In Figure S3 and S4, please use multiple adjustment p values to determine significance. Additionally, more information about these reverse mutants compared to the WT can be included as notes in the legend for better understanding and comparisons.

The data for Figures S3 and S4 are discussed in detail within the text on lines 299-308. The data in all the left-hand panels (b-arrestin recruitment) do not require statistical analyses because there was no response with the reverse mutants. All of the data in the right-hand panels (cAMP) were analyzed statistically and there were no differences between the mutant and WT results, except for the Rev S7-S10 and Rev 1718 mutants which exhibited a ~2-fold (p < 0.05) increase each in the DA EC50 value (Figure S3 and S4). This is not an adjusted p value, as only a single comparison was made against WT. We do not believe this small result to be biologically meaningful, however.

(9)   At lines 237-255, please add the statistical significance p value for each comparison. For other comparisons in the main test, please include significance p values accordingly.

All p values have been added to the text as requested and elsewhere when stating statistical comparisons.